# Goal-Conditioned Reinforcement Learning with Virtual Experiences

## Abstract

Goal-conditioned reinforcement learning often employs a technique known as *Hindsight Experience Replay* (HER) for data augmentation by relabeling goals. However, HER limits goal relabeling to a single trajectory, which hinders the utilization of experiences from diverse trajectories. To address this issue, we present a curriculum learning method to construct virtual experiences, incorporating actual state transitions and virtual goals selected from the replay buffer. Considering that virtual experiences may contain a lot of noise, we also propose a self-supervised subgoal planning method that guides the learning of virtual experiences by imitating the subgoal-conditioned policy. Our intuition is that achieving a virtual goal may be challenging for the goal-conditioned policy, whereas simplified subgoals can provide effective guidance. We empirically show that the virtual experiences from diverse historical trajectories significantly boost the sample-efficiency compared to the existing goal-conditioned reinforcement learning and hierarchical reinforcement learning methods, even enabling the agent to learn tasks it has never experienced.[1]

## 1 Introduction

Goal-conditioned reinforcement learning (GCRL) is an advanced form of deep reinforcement learning (RL) that aims to find an optimal solution for tasks involving multiple goals simultaneously and requiring long-term decision-making. For instance, it can effectively guide a multi-legged robot to a predetermined location on a map (Paul et al., 2019) or skillfully manipulate a robotic arm to grasp objects on a platform (Zhang et al., 2020). Unlike many existing methods that necessitate intricate reward functions tailored to each task (Ng et al., 1999; Brys et al., 2014; Devidze et al., 2022), GCRL simplifies the problem's complexity by relying solely on binary reward signals that indicate goal achievement (Nair et al., 2018; Liu et al., 2022). Nevertheless, binary rewards introduce the challenge of reward sparsity.

Humans possess the ability to synthesize past experiences and utilize this knowledge to efficiently adapt to new tasks. Building upon this concept, HER (Andrychowicz et al., 2017) introduces a method to reconstruct data from unsuccessful past experiences, partially mitigates the challenge of sparse rewards by selecting the actual reached states in the trajectory as the relabeled goals. However, HER requires limiting the relabeled goals and state transitions within a single task trajectory, which is not conducive to integrating experience from different tasks. For example, once the skills of "grabbing apples" and "opening a drawer" are mastered, there is no need to relearn them from scratch when performing the task of "putting apples into a drawer"; instead, these skills can be combined.

As demonstrated in HER, merely broadening the selection scope of relabeled goals to the entire replay buffer yields no benefit. An intuitive explanation for this is that, without real trajectory support, the connection between actual state transitions and virtual goals remains tenuous, injecting substantial noise into the reconstructed data. During policy learning, such data not only fail to provide valuable experience but also undermine learning stability. To address this issue, we introduce curriculum learning when selecting virtual goals and use subgoals to simplify these goals. Inspired by prior work (Pitis et al., 2020), we propose a method for selecting virtual goals based on task

---

[1] Code is available at https://anonymous.4open.science/r/VE-7224/

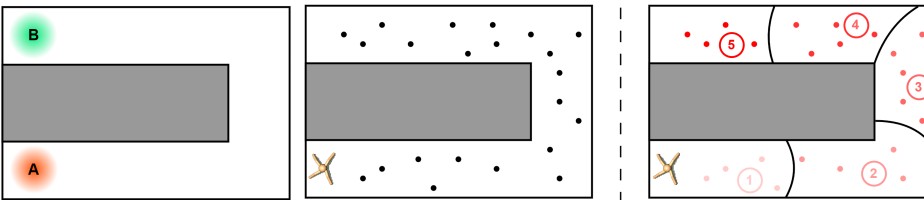

(a) The high state probability density near $s_A$ and $s_B$ does not reflect the reachability of task $(s_A, s_B)$.

(b) The left shows random sampling of states, relabeled as goals, from the replay buffer. The right demonstrates filtering these relabeled goals $g$ by increasing difficulty, determined by task probability density $e(s_A, g)$ (e.g. selecting ① to ⑤ in training progresses).

Figure 1: Illustration of (a) the disadvantage of state probability density and (b) expanding the sampling range to the replay buffer versus our approach for selecting by task probability density in the U-shape AntMaze environment (AntMaze-U).

probability density. This approach aims to provide virtual goals of increasing difficulty, thereby reducing subgoal planning complexity and improving the accuracy of subgoal-conditioned policies. Furthermore, we propose a simple self-supervised subgoal planning method that differs from previous studies (see Section 2 for details). This method, similar to high-level policies in hierarchical RL, leverages sampled self-supervised data in place of traditional reward signals to learn. Our key contributions are as follows:

- **Task-probability-density-based curriculum learning:** initially, we expand the selection scope of relabeled goals to the entire replay buffer, including not only actual goals within the same trajectory but also virtual goals across different trajectories. Subsequently, we envisage that gradual task difficulty facilitates the learning of long-term goals and improves the precision of subgoal planning. Hence, we propose using task probability density rather than state probability density to monitor an agent's learning progress and use it as a reference for selecting virtual goals. Implementing curriculum learning based on task probability density can further enhance the success rate of achieving multiple goals.

- **Self-supervised subgoal planning:** to effectively learn from virtual experiences, we propose a self-supervised subgoal planning method. Simplified subgoals reduce the task's complexity, and subgoal-conditioned policies can serve as reasonable imitation targets for goal-conditioned policies, thereby constraining and guiding the policy improvement process.

In summary, we propose a general framework for utilizing virtual experience (VE) to improve sample-efficiency in GCRL, and demonstrate its performance over existing GCRL and hierarchical RL methods in various robot navigation and control tasks based on the MuJoCo simulator (Todorov et al., 2012). The experimental results indicate that the proper use of virtual experience can significantly accelerate the learning of multiple goals. Moreover, when combined with subgoal planning, it provides valuable guidance for achieving complex long-term goals. This approach can even assist agents in accomplishing tasks they have never encountered before.

## 2 RELATED WORK

**Goal relabeling.** HER demonstrates that off-policy deep RL algorithms can reuse data in the replay buffer by relabeling goals in episodic trajectories. It is widely utilized as a core component in many related studies (Levy et al., 2017; Ren et al., 2019; Pitis et al., 2020; Durugkar et al., 2021; Kim et al., 2023). A growing consensus suggests that HER can be viewed as implicit curriculum learning, structuring curricula within historical trajectories by using transition data. CHER (Fang et al., 2019) optimizes the curriculum by considering the similarity and diversity of relabeled goals, but still does not overcome the limitations of a single trajectory. (Kuang et al., 2020) and (Pong et al., 2019; Pitis et al., 2020) also use the density estimation model to aid in goal selection. By prioritizing low-density goals based on the density estimation of the achieved goals, they increase the likelihood of these goals becoming candidates, thus facilitating curriculum learning.

These approaches are similar to ours, but they assume that, under the premise of a fixed initial state, a visited state is regarded as the achieved goal. However, when the initial state distribution encompasses the entire state space, a visited state cannot reliably indicate that the goal has been achieved, as a complete task is defined by both the initial state and the goal (see Figure 1 (a)). Our idea is to estimate the task probability density to track the learning progress of the agent, which is also applicable to the case of a fixed initial state.

**Subgoal planning in GCRL.** The curriculum learning mentioned above can be regarded as planning within exploration or learning process. Planning is also a crucial concept in hierarchical RL. When confronted with a long-horizon problem, high-level policies plan with temporally extended actions, known as abstract actions, directing the low-level policies to execute primitive actions to complete the interaction (Bacon et al., 2017; Park et al., 2024). In the context of GCRL, these abstract actions can be considered as subgoals employed as targets for the low-level policies. Current approaches to subgoal planning can primarily be categorized into learning-based and search-based methods. (Badrinath et al., 2024) treats the midpoint of the path as a subgoal in the context of offline RL and trains a subgoal prediction module using a transformer. (Jurgenson et al., 2020) establishes a connection between subgoal planning and the dynamic programming equation, which inherently benefits navigation and obstacle avoidance tasks. However, extending this approach to high-dimensional goal spaces remains challenging. These learning-based methods generally require only a single forward inference to accomplish subgoal planning, making them more efficient than search-based methods. But they also encounter challenges of learning. In contrast, VE can accurately predict subgoals based only on sampled self-supervised signals. Additionally, it is a common practice to apply search-based subgoal planning in graphs (Huang et al., 2019; Eysenbach et al., 2019; Hoang et al., 2021; Zhang et al., 2021; Kim et al., 2021; 2023). Specifically, these methods construct a graph where nodes and edges correspond to states and inter-state distances respectively, and various historical experiences are linked together by the graph. The purpose of planning is to find the shortest path consisting of nodes to reach the goal, where the nodes can be regarded as the subgoals. To apply graph-based planning to complex environments, some work focuses on learning robust representations that cover the entire state space. Under the constraint of a limited number of nodes, (Hoang et al., 2021) measures the similarity between subsequent features to ensure that the nodes cover as large an area as possible. (Zhang et al., 2021) learns a latent space and obtains graph nodes through clustering. The adjacent nodes reflect the temporal reachability. Others consider combining traditional path planning methods, such as farthest point sampling (Huang et al., 2019), to find shortest paths based on the graph (Kim et al., 2021; 2023).

Although graphs reducing the complexity of planning compared to state space, they still face challenges related to computational complexity and suboptimality. As PIG (Kim et al., 2023) demonstrates, searching for shortest paths for tasks in each batch of training data significantly increases computational overhead, limiting the batch size to a small value. Moreover, a finite number of nodes cannot plan optimal paths for all tasks. In contrast, VE is not subject to this limitation. It's similar in form to the high-level policy in hierarchical RL, but it does not control agent exploration; it is solely used to guide policy learning.

# 3 PRELIMINARY

## 3.1 GOAL-CONDITIONED REINFORCEMENT LEARNING

GCRL aims to learn policies that can achieve multiple goals. Our reinforcement learning agent interacts with a discounted, infinite-horizon, goal-conditioned Markov decision process (MDP) (Sutton, 2018), defined by the tuple $(\mathcal{S}, \mathcal{A}, \mathcal{G}, p, r, \gamma)$. $\mathcal{S}$, $\mathcal{A}$, and $\mathcal{G}$ are the state, action and goal spaces, respectively, and $\gamma \in (0, 1)$ is the discount factor. The transition function is denoted as $p(s_{t+1}|s_t, a_t)$ where $s_t, s_{t+1} \in \mathcal{S}$ and $a_t \in \mathcal{A}$, and the reward function as $r(s_t, a_t, g)$. We set the reward $r$ to $-1$ for all actions until the agent reaches the goal. With the goal distribution $\rho(g)$, the objective of an agent is to find the optimal policy $\pi^*(\cdot|s, g)$ that maximizes the expected sum of discounted rewards in $T$ steps

$$J(\pi) = \mathbb{E}_{g \sim \rho(g), s_t \sim p^\pi(\cdot|s_0, g)} \left[ \sum_{t=0}^{T} \gamma^t r(s_t, a_t, g) \right] \qquad (1)$$

We follow the standard off-policy actor-critic paradigm (Silver et al., 2014; Mnih et al., 2016; Fujimoto et al., 2018). Specifically, we sample a batch of data from the replay buffer $\mathcal{B}$. In the policy evaluation phase, an goal-conditioned action-value function (critic) $Q(s, a, g)$ (Schaul et al., 2015) with parameters $\beta$ gets update by minimizing the Bellman error

$$Q_{\beta_{k+1}} = \arg\min_{\beta} \mathbb{E}_{(s_t, a_t, s_{t+1}, g) \sim \mathcal{B}} \left[ y_t - Q_{\beta_k}(s_t, a_t, g) \right]^2 \tag{2}$$

with the target value $y_t = r(s_t, a_t, g) + \gamma \mathbb{E}_{a_{t+1} \sim \pi(\cdot|s_{t+1}, g)} [Q_{\beta_k}(s_{t+1}, a_{t+1}, g)]$.

During the policy improvement phase, the policy (actor) $\pi$ with parameters $\theta$ is updated in the direction that maximizes the expected value of current critic $Q$.

$$\pi_{\theta_{k+1}} = \arg\max_{\theta} \mathbb{E}_{(s, g) \sim \mathcal{B}} \left[ Q\left(s, \pi_{\theta_k}(a|s, g), g\right) \right] \tag{3}$$

### 3.2 GOAL RELABELING

In this paper, we assume states and goals share the same data space denoted by $\mathcal{S} = \mathcal{G}$, allowing each state to be treated as a goal. Typically, historical data is stored in a buffer as trajectories $\mathcal{B} = \{\tau_1, \tau_2, ...\}$, where each trajectory $\tau_i = \{s_0^i, a_0^i, s_1^i, a_1^i, ..., s_T^i, g^i\}$ represents the exploration history of a task $(s_0^i, g^i)$. We formalize the go relabeling method in Hindsight Experience Replay (Andrychowicz et al., 2017) as follows

$$(s_t^i, a_t^i, s_{t+1}^i, g^i, r(s_{t+1}^i, g^i)) \rightarrow (s_t^i, a_t^i, s_{t+1}^i, \boldsymbol{s_{t+n}^i}, r(s_{t+1}^i, \boldsymbol{s_{t+n}^i})) \tag{4}$$

where $r(s_t^i, a_t, g^i)$ is simplified as $r(s_{t+1}^i, g^i)$ and $n \in [1, T - t]$. This implies that we can use future states $s_{t+n}^i$ or final states $s_T^i$ within the same trajectory as relabeled goals and recalculate the rewards $r$. These reconstructed data allow any task from that trajectory to have a chance to receive a positive reward signal.

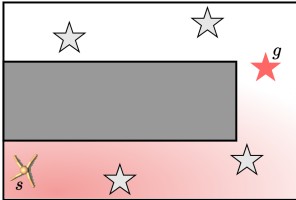 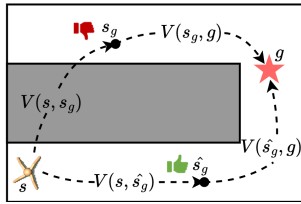 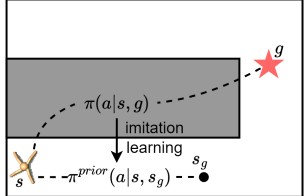

Figure 2: VE consists of three parts. **Left:** the red gradient area represents the goal range learned from state $s$, recorded by task probability density estimation. We select the goal on the boundary as the relabeled goal. **Middle:** it illustrates the learning process of self-supervised subgoal planning. When the subgoal $s_g$ generated by the high-level policy $\pi^h$ is not better than the sampled subgoal $\hat{s}_g$, the probability of the sampled subgoal $\pi^h(\hat{s}_g|s, g)$ is increased. **Right:** during policy improvement, the goal-conditioned policy is required to imitate the subgoal-conditioned policy.

## 4 LEARNING GOAL-CONDITIONED POLICY WITH VIRTUAL EXPERIENCES

In this section, we introduce a new framework named VE, which consists of three main parts: (1) filtering relabeled goals by task probability density for building a step-by-step curriculum; (2) a subgoal planning method based on self-supervised learning for simplifying the relabeled goals; and (3) taking subgoal-conditioned policies as imitation targets to accelerate policy improvement. We provide an illustration of our framework base on AntMaze-U in Figure 2.

### 4.1 RELABELING GOALS TO CONSTRUCT VIRTUAL EXPERIENCE

As introduced in Section 3.2, the scope of goal relabeling in HER is limited. Our idea is that any goal can be used for relabeling when reconstructing data, such as the goal sampled from replay buffer or designed by a human. We term this approach *generalized goal relabeling* and present it as a more generalized form compared to Formula (4):

$$(s_t^i, a_t^i, s_{t+1}^i, g^i, r(s_{t+1}^i, g^i)) \rightarrow (s_t^i, a_t^i, s_{t+1}^i, g^j, r(s_{t+1}^i, g^j)) \tag{5}$$

In this paper, we discuss the relabeled goal $g^j$ sampled from the states of replay buffer. If $j = i$, it is equivalent to HER, and we refer to these goals as *actual goals*. Otherwise, if $j \neq i$, these are termed *virtual goals*. Intuitively, a curriculum that gradually increases in difficulty can help an agent complete tasks faster. As shown in the Figure 2 (**Left**), given a state $s$, we hope that the virtual goal $g$ in the virtual experience is on the boundary of the reachable state from $s$. In order to effectively evaluate the reachability of the task $(s, g)$, we propose estimating the task probability density $e$ in curriculum, acting as a filter to screen suitable virtual goals from random sampling.

Specifically, we establish a dedicated buffer $\mathcal{B}_l$ to record the learned data. Inspired by previous work (Pitis et al., 2020), we approximate the learning frequency of tasks during training as their probability density. When screening the virtual goal for a state $s$, our initial step is to sample a set of states from the replay buffer $\mathcal{B}$ which serve as candidate virtual goals $g$. Subsequently, we compute their task probability density $e(s, g)$ and select one that satisfies conditions (e.g. $0.8\bar{e} < e(s, g) < 1.2\bar{e}$, with $\bar{e}$ denoting the mean) as the virtual goal. It is crucial to note that our method of employing a fixed range to filter out virtual goals on the boundary is empirical and performs well in simple environments (see Figure 5). This process will benefit from future improvements. For details on the learning process of task probability densities, please refer to Appendix A.

A significant advantage of virtual experience is that it broadens the distribution of tasks in the reconstructed data, particularly for tasks that are difficult to achieve through traditional exploration. Learning these tasks through virtual experience, however, is inherently challenging. Firstly, from a policy perspective, there is a lack of connection between the original state transitions $(s_t^i, a_t^i, s_{t+1}^i)$ and the relabeled goals $g^j$. This reduces the accuracy of policy evaluation. Secondly, policy improvements based on virtual experience are also inefficient. So we simplify the virtual goal through the subgoal planning method in Section 4.2.

## 4.2 SELF-SUPERVISED SUBGOAL PALNNING

We implement subgoal planning through a high-level policy $\pi^h(s, g, k)$ similar to that used in hierarchical RL. It is important to emphasize that subgoals are only utilized for policy improvement and do not guide exploration. Consistent with previous work, we consider the state value function $|V(s, g)|$ as the discounted distance from the state $s$ to the goal $g$. Based on this, we define the transfer distance vector $v(s, \{s_g\}_k, g)$ using the subgoal sequence $\{s_g\}_k = \{s_{g_1}, s_{g_2}, .., s_{g_k}\}$ as:

$$v(s, \{s_g\}_k, g) = \left[ |V(s, s_{g_1})| , |V(s_{g_1}, s_{g_2})| , \cdots , |V(s_{g_{k-1}}, s_{g_k})| , |V(s_{g_k}, g)| \right]^\top \quad (6)$$

The high-level policy $\pi^h$ with parameters $\phi$ is trained to predict optimal subgoals that minimize the $p$-norm of the transition distance vector with loss

$$\mathcal{L}_{\pi^h}(\phi) = \mathbb{E}_{(s,g) \sim \mathcal{B}, s_{g_k} \sim \pi^h(\cdot|s,g,k)}[d(s, \{s_g\}_k, g)] \quad (7)$$

where $d(s, \{s_g\}_k, g) = \|v(s, \{s_g\}_k, g)\|_p$.

The novelty of our approach lies in applying the idea of Advantage-Weighted Regression (AWR) (Peng et al., 2019) to the learning of high-level policies. Specifically, we equate the self-supervised signal $d(s, \{\hat{s_g}\}_k, g)$ to the value $Q$ in the advantage function $A = Q - V$, where the prediction of the high-level policy $d(s, \{s_g\}_k, g)$ corresponds to the value $V$. The difference is that we compare the advantages of sampled subgoals over those output by high-level policies, rather than comparing the advantage of the current policy over the historical (or mixed) policy as in AWR. Consequently, we propose randomly sampling states from the replay buffer as subgoals $\{\hat{s_g}\}_k$. If the advantage $A > 0$, it indicates that the prediction of the high-level policy is inferior to the sampled subgoals and should be adjusted to approximate them. Conversely, if the advantage $A < 0$, the prediction of high-level policy should deviate from the sampled subgoals. The gradient of loss $\mathcal{L}_{\pi^h}(\phi)$ is

$$\nabla_\phi \mathcal{L}_{\pi^h}(\phi) = \nabla_{\phi \{\hat{s_g}\}_k \sim \mathcal{B}, s_{\hat{g_k}} \sim \{\hat{s_g}\}} \log \pi_\phi^h(s_{\hat{g_k}}|s, g) A^{\pi^h}(s, g) \quad (8)$$

where $A^{\pi^h}(s, g) = d_{\{\hat{s_g}\}_k \sim \mathcal{B}}(s, \{\hat{s_g}\}_k, g) - d_{\{s_g\}_k \sim \pi^h}(s, \{s_g\}_k, g)$. We will introduce methods for choosing the number of subgoals $k$ and the $p$-norm in the Appendix C.

## 4.3 SUBGOAL-CONDITIONED IMITATION LEARNING

In this paper, we consider the control problem in continuous space, so we assume that the action follows a Gaussian distribution. The process of the goal-conditioned policy imitating the subgoal-

(a) AntMaze-U (b) AntMaze-S (c) AntMaze-Π (d) AntMaze-W    (e) Sawyer    (f) Reacher

Figure 3: Visualizations of our experimental environments, where the Sawyer is a pixel-based task. When trained in AntMaze, an agent starts at a random point, and aims to reach a random goal. The most challenging tasks used for evaluation in AntMaze: the blue point and the position of an ant indicates the goal and the initial point, respectively.

conditioned policy can be written as $D_{\mathrm{KL}}(\pi^{prior}(s, s_{g_k})||\pi(s, g))$. Additionally, the optimization objective of imitation learning can be weighted into the policy improvement as a regularization term

$$\pi_{\theta_{m+1}} = \arg\max_{\theta} \mathbb{E}_{(s,g)\sim\mathcal{B}, s_{g_k}\sim\pi^h} \left[ Q\left(s, \pi_{\theta_m}(s, g), g\right) - \frac{\alpha}{k} \sum_{i=1}^{k} D_{\mathrm{KL}}(\pi^{prior}(s, s_{g_i})||\pi_{\theta_m}(s, g)) \right]$$

(9)

By generating subgoal-conditioned policies based on simplified subgoals, we recognize the key role of prior policies $\pi^{prior}$ in imitation learning. Previous methods utilize historical (Pertsch et al., 2021), expert (Sonabend et al., 2020), or current policies (Kim et al., 2023) as prior policies, while we conduct a comparison and select the soft-updated historical policies (see Figure 7 **right** for more details). This selection stems from the reason that expert policies are usually difficult to obtain, historical policies lag in acquiring knowledge, and current policies may lack stability. In contrast, the soft-updated historical policy strikes a good balance between historical and current policies. The full algorithm is summarized in the Appendix B.

## 5 EXPERIMENTS

Our experiments aim to answer the following questions: **(1)** Can VE improve sample-efficiency and performance in continuous control tasks compared to the baseline? **(2)** Is curriculum learning based on task probability density effective? **(3)** What impact does the proportion of virtual experiences in the training data have on the policy? **(4)** How does imitating subgoal-conditioned policies influence goal-conditioned policy learning? **5** Can VE integrate knowledge from different tasks?

### 5.1 EXPERIMENTAL SETUP

**Environments.** Following the experimental setup of previous research (Kim et al., 2023), we design a series of challenging environments based on MuJoCo simulator (Todorov et al., 2012). Specifically, our navigation task consists of {U, S, Π, W}-shaped AntMaze environments, and our robotic arm control tasks involve Sawyer and Reacher (see Figure 3 for the visualization of environments), with Sawyer being a pixel-based task. In contrast to (Huang et al., 2019; Kim et al., 2023), where the goal space is defined on the 2-dimension that represents the $(x, y)$ position of agent in AntMaze tasks, we use a entire 31-dimensional state space as the goal space following the setting of RIS (Chane-Sane et al., 2021). In the Reacher task, the goal space is 3-dimension. The configuration of the Sawyer task remains consistent with (Chane-Sane et al., 2021), where we obtain a pixel representation of the target through pre-sampling. We provide more details in Appendix F.

**Implementation.** VE is based on the standard Actor-Critic framework. In order to reduce the impact of $Q$ value overestimation, we use double $Q$ value function. During the network update, we set the target network for the actor and the critic, and adopt a soft update method with the coefficient 5e-3. We use the actor target policy as the prior policy, which can effectively ensure the stable convergence of the policy in our experiments. For the baseline, we follow the original setup (see more details in Appendix G).

**Evaluation.** Our experimental settings for navigation include random start points and random endpoints, requiring the agent to master the entire map. This setup is more challenging than the one with

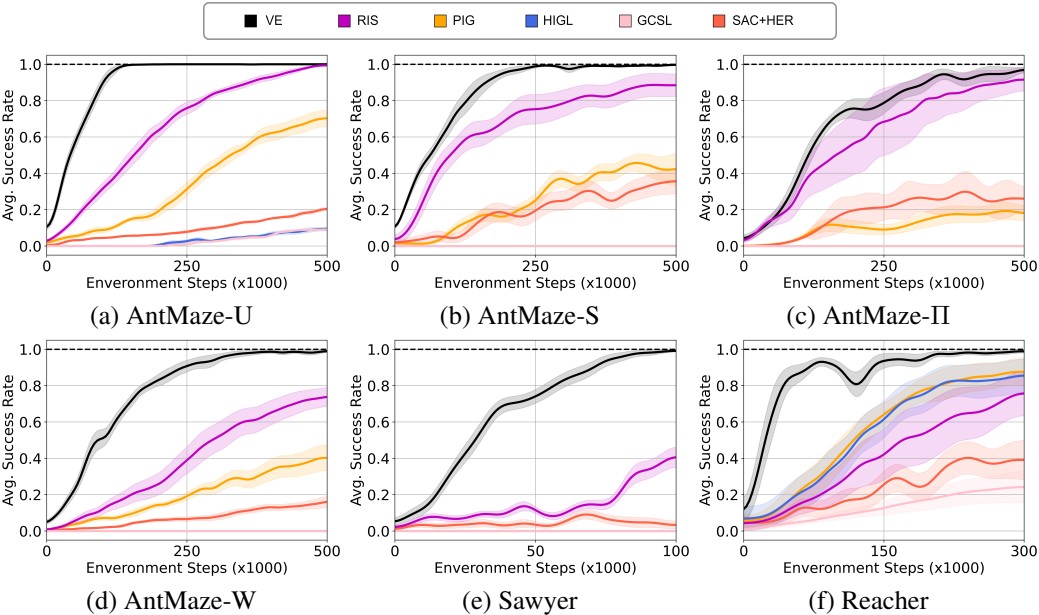

Figure 4: Learning curves for agents in different environments. All methods are run with 4 seeds, and the solid and shaded lines represent mean and standard deviation, respectively. It is important to note that PIG, HIGL and GCSL cannot handle pixel-based tasks (Sawyer), and both HIGL and GCSL are nearly unsuccessful in complex navigation tasks.

fixed start and end points. Additionally, for each navigation map, we define the most difficult task as the farthest path between the origin and destination (see Figure 3). Test results for the random task are reported using four different seeds, with 100 test episodes conducted every 5,000 training steps. For visual clarity, we smooth all the curves equally.

**Baselines.** We select several representative methods as comparison baselines (see the Appendix H for detailed hyperparameter settings). **PIG** (Kim et al., 2023): A subgoal self-imitation framework based on graph planning, where subgoals are planned on the graph using a skipping mechanism. The version combined with MSS (Huang et al., 2019) is utilized in the experiment; **HIGL** (Kim et al., 2021): A hierarchical RL algorithm combined with path planning. It maintains a queue of subgoals by computing novelty and coverage, selecting appropriate subgoals for the low-level policy through path planning; **RIS** (Chane-Sane et al., 2021): A method that predicts the midpoint of a path as a subgoal, and also uses the subgoal to guide the current policy learning; **GCSL** (Ghosh et al., 2019): An imitation learning method for generating supervised data by relabeling trajectories. It achieves further goals based on the optimal substructure of GCRL by continuously imitating successful trajectories; **SAC** (Haarnoja et al., 2018): A classic RL algorithm based on maximum entropy, and it has a good performance in continuous control tasks. We combine SAC and HER to adapt to GCRL.

## 5.2 RESULTS

In Figure 4, our proposed VE demonstrates a substantial performance advantage over other methods across various control tasks. Methods such as PIG and HIGL, which integrate historical experience by building a graph, require a planning strategy or high-level policy when testing. In contrast, our method relies solely on goal-conditioned policy. Furthermore, building a graph model increases the computational complexity, causing PIG and HIGL need more training steps to achieve optimal performance, as described in the original results (see (Kim et al., 2023)).

Additionally, it should be emphasized that, except for VE and RIS, the goal space used by other methods includes only coordinate positions in the AntMaze environments. This simplification significantly reduces the complexity of subgoal planning and the learning of goal-conditioned policies. This also highlights the advantage of VE in handling high-dimensional goal spaces. Compared to

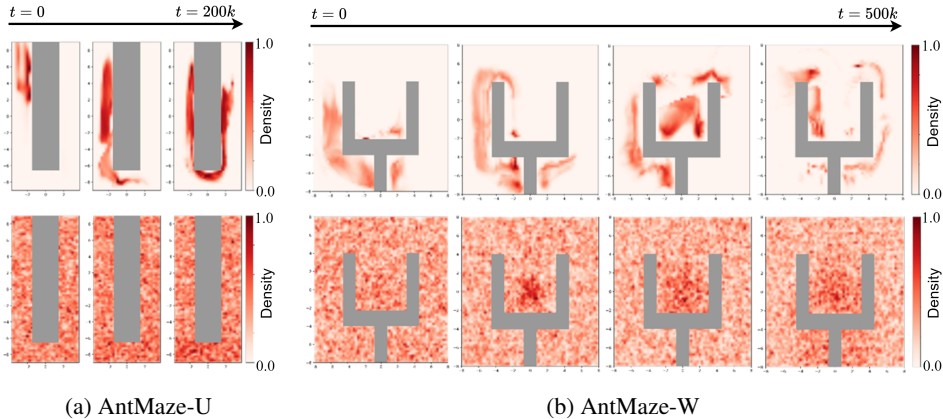

(a) AntMaze-U                  (b) AntMaze-W

Figure 5: We visualize the comparison between task probability density (up) and state probability density (down) during the training process in AntMaze-U and AntMaze-W (the state $s$ in $e(s, g)$ is chosen to be located as shown in Figure 3, and the values of $e$ are normalized in each subfigure), illustrating the curriculum learning composed of virtual goals with gradually increasing difficulty.

RIS, our subgoal planning demonstrates greater accuracy (see Appendix D.1). Furthermore, the efficiency of VE in learning multiple tasks has improved due to curriculum learning, which screens suitable virtual goals. In contrast to GCSL, which merely imitates historical trajectories, VE enhances training data by incorporating virtual experiences, thereby achieving higher sample utilization. Since our method ultimately focuses on imitation learning in Equation (9), SAC+HER can be regarded as a baseline without imitation learning. From the results, we can see that SAC+HER cannot learn reliable policies. Then, we evaluate the performance of VE and RIS on the most difficult tasks (see Figure 6). This implies that the application of virtual transition experiences can be highly beneficial in overcoming challenging tasks. Additional results are available in the Appendix D.2.

## 5.3 ABLATION STUDIES

In this section, we conduct ablation validation experiments on various key components of VE to demonstrate the effectiveness of our method. All experiments were carried out in U-shaped and W-shaped maps.

**Goal relabeling with task probability density.** Goal relabeling is a crucial method for constructing virtual experiences. We experimentally verify the impact of different ratios of actual and virtual experiences on VE. The results, as shown in Figure 7 (left), indicate that the optimal ratio is "a:v=0.5:0.5". Notably, when relying solely on actual experiences ("a:v=1.0:0"), the policy performs comparably to SAC+HER, demonstrating our method's effective utilization of these samples for learning. In contrast, relying exclusively on virtual experiences ("a:v=0:1.0") results in the policy failing to learn effectively. We attribute this to actual experiences enabling the agent to quickly master simple tasks, while virutal experiences help the agent tackle more challenging, long-horizon tasks. When the agent has not yet learned simple tasks, it cannot acquire knowledge from virtual experience. Furthermore, we compare the effects of screening virtual

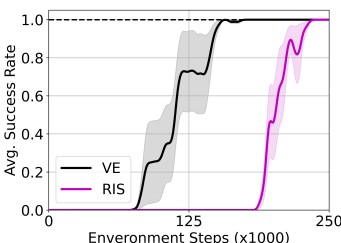

Figure 6: Average success rate of VE and RIS in the most difficult task on AntMaze-U.

targets based on task probability density versus randomly sampling virtual goals. The Figure 7 (middle) shows that providing virtual goals with gradually increasing difficulty in course learning can help further improve learning efficiency. The visualization result of task probability density is shown in Figure 5. The state probability density (down) can only assess the frequency of visiting a state, not the reachability of task. In contrast, the task probability density (up) can monitor the agent's learning progress and offer curriculum of gradually increasing difficulty during training.

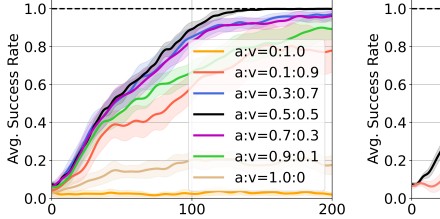 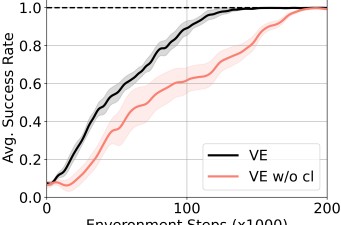 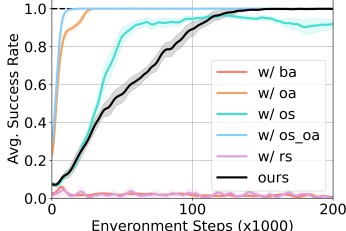

Figure 7: We investigate (**left**) the impact of different ratios of actual goals (a) and virtual goals (V) on VE, (**middle**) the impact of curriculum learning (cl) on VE, and (**right**) the impact of different subgoal strategies and prior strategies on VE. Here, "os" denotes oracle subgoals, "oa" denotes oracle actions, "ba" denotes behavioral actions, and "rs" denotes random subgoals. All the above experiments are tested in environment AntMaze-U.

**Self-Supervised Subgoal Planning.** We devise two alternatives to demonstrate the effectiveness of our subgoal planning model (see Figure 7 (right)): sampling subgoals by method "os" or "rs." It is evident that more precise subgoals can significantly expedite the learning of goal-conditioned policies. We also observe that the oracle subgoal does not demonstrate substantial improvement during the initial stages of training. Our assumption is that, at this stage, the learning task is fundamentally simple, causing the high-level policy planning subgoals to align closely with the oracle subgoals. We visualize the subgoals of the testing phase, with more details provided in the Appendix D.3.

**Subgoal-Conditioned Imitation Learning.** The core idea of our method of using virtual experience is to imitate the subgoal-conditioned policy. Combined with the Equation 9, our method directly applies the knowledge contained in the virtual experience to the policy instead of the value function. Intuitively, this approach is expected to be more efficient than first optimizing the value function and subsequently improving the policy by maximizing the value function. To further verify the exceptional capability of the VE in integrating historical experience, we design an experiment on AntMaze-U. During training, we constrain the range of random task generation so that the initial state and the goal appear only on the same side of the U-shaped map. We evaluate the average success rate on both random tasks and the most difficult tasks, reflecting the VE's ability to inte-

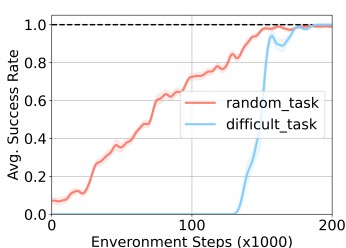

Figure 8: Average success rate of VE on constrained AntMaze-U.

grate knowledge learned on both sides of the map. As shown in Figure 8, even under task constraints, the VE successfully solves all navigation tasks on AntMaze-U, including the most challenging ones, with only slightly reduced learning efficiency compared to unrestricted conditions.

## 6 CONCLUTSION

**Limitations and Future Work.** While VE has shown promising results, it is essential to acknowledge certain limitations. For instance, the hyperparameter $k$ is related to the task's scale. For more complex environments, accurate subgoals with different decision horizons can help further improve the quality of subgoal-conditioned policies. At the same time, complex environments also introduce challenges in estimating task probability density. Future research can consider applying subgoals to both agent exploration and policy learning. Moreover, filtering low-quality subgoal-conditioned policies or considering safety constraints in subgoal planning (García & Fernández, 2015; Gu et al., 2022) represents potential directions for future investigation.

We propose a goal-conditioned reinforcement learning method, VE, that integrates knowledge from different historical tasks by constructing virtual experiences through a curriculum. We highlight that simply using virtual experiences does not effectively aid policy learning, whereas the self-supervised subgoal planning we propose significantly reduces the difficulty of achieving virtual goals. By imitating subgoal-conditioned policies, the agent is expected to overcome the exploration dilemma and accomplish tasks that require complex long-term decisions.

## 7 REPRODUCIBILITY STATEMENT

We provide the implementation details of our method in Section 5.1 and Appendix G. We also open-source our codebase.

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

## A  TASK PROBABILITY DENSITY

In order to track the agent's progress in learning different tasks, we approximate the training frequency of the task as the task probability density $e$. This is based on the simple intuition that the more frequently a task is trained, the higher its probability of being sampled. Given an initial state, we then select one from candidate virtual goals that aligns with the current learning progress frontier to guide the learning process. Specifically, we compare two commonly used probability density estimation methods, KDE (Kim & Scott, 2012) and Flow (Papamakarios et al., 2021). Our experiments found that the Flow model had better performance in high-dimensional space. At the same time, the Flow model based on RealNVP (Dinh et al., 2016) was more efficient in the calculation, and its time consumption was significantly reduced compared to the KDE method. Therefore, we implemented the Flow model based on RealNVP to estimate task probability density. We designate a dedicated replay buffer $\mathcal{B}_l$ for recording recent training data. The Flow model is updated in parallel with the policy learning process to effectively track learning progress.

## B  ALGORITHM

We provide algorithm that represent VE in Algorithm 1.

---

**Algorithm 1** VE

---

Initialize replay buffer $\mathcal{B}, \mathcal{B}_l$
Initialize $Q_\beta, \pi_\theta, \pi_\phi^h$
Initialize task probability density model (Flow model based on RealNVP) $e$
  1: **for** $k = 1, 2, ...$ **do**
  2:    Collect experience in $\mathcal{B}$ using $\pi_\theta$
  3:    Sample batch data $d \sim \mathcal{B}$
  4:    Go relabeling with task probability density model $e$
  5:    Store batch data $d$ in $\mathcal{B}_l$
  6:    Update task probability density model $e$ with data from buffer $\mathcal{B}_l$
  7:    Sample batch state $s \sim \mathcal{B}$ as subgoal baseline $\hat{s_g}$
  8:    Planning subgoals with $\pi_\phi^h$
  9:    Update $Q_\beta$ using Equation equation 2 (*Policy Evaluation*)
 10:    Update $\pi_\phi^h$ using Equation equation 8 (*Self-Supervised Subgoal Planning*)
 11:    Update $\pi_\theta$ using Equation equation 9 (*Policy Improvement with Subgoal-Conditioned Imitation Learning*)
 12: **end for**

---

## C  SELF-SUPERVISED SUBGOAL PLANNING

In Section 4.2, we discuss how to perform subgoal planning via self-supervised learning. There are two key parameters, the norm $p$ and the number of subgoals $k$.

First, we demonstrate that under the definition of reward function in Section 3.1, using the 1-norm leads to subgoal degradation, whereas using the 2-norm and the $\infty$-norm can theoretically learn the optimal subgoal estimate.

Obviously, according to the definition of value function ($r = -1, \gamma = 0.99$), we can get

$$V = r + \gamma r + \gamma^2 r + \cdots + \gamma^T r \qquad (10)$$
$$= -(1 - \gamma^T)(\text{omit coefficient})$$

and based on Equation 6, we can get

$$d = \|v(s, \{s_g\}_k, g)\|_p \tag{11}$$

$$= (\sum_{i=1}^{k} |V(s_{g_i})|^p)^{\frac{1}{p}}$$

$$= (\sum_{i=1}^{k} (1 - \gamma^{t_i})^p)^{\frac{1}{p}}$$

Since $f(x) = x^p$ $(x \in (0,1), p \geq 1)$ is a convex function when $p \geq 2$, let $x_i = 1 - \gamma^{t_i}$ and $\sum t_i = T$. Using Jensen's inequality, we know that when $p = 1$, the minimum value of $L$ is obtained at $t_1 = t_2 = \cdots = t_{k-1} = 1$ and $t_k = T - k + 1$. This means that the number of $k - 1$ subgoals degenerate into adjacent states, which is not conducive to high-level policy learning. And when $p \geq 2$, the minimum value of $d$ is obtained at $t_1 = t_2 = \cdots = t_k = T/k$. This is equivalent to equally partitioning the optimal path from the state $s$ to the goal $g$ among the $k$ subgoals. We experimentally compared the results for $p = \{1, 2, \infty\}$ and $k = \{1, 2\}$ on AntMaze environments in Figure 9.

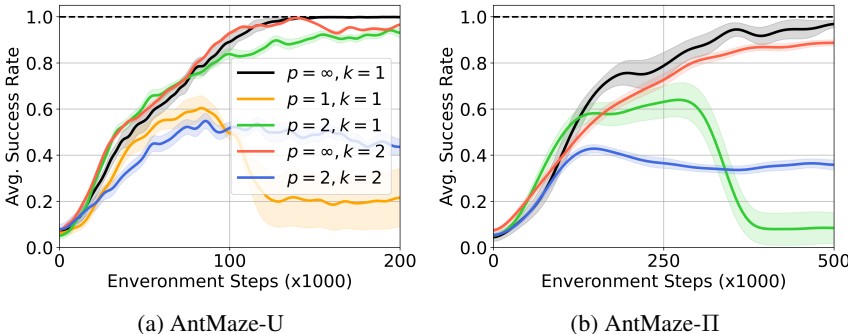

(a) AntMaze-U            (b) AntMaze-Π

Figure 9: Learning curves for agents in (a) AntMaze-U and (b) AntMaze-Π with different hyperparameters $p$ and $k$.

We firstly observe that when $p = 1$, the agent is incapable of learning strategies to address a variety of tasks, which aligns with our theoretical proof. Although the inference indicates that when $p = 2$ or $p = \infty$, the high-level policy can formulate accurate subgoals, the empirical evidence suggests that a degradation of the goal-conditioned policy occurs when $p = 2$. We speculate that this occurrence might be due to the necessity for the sampled subgoals to secure an overall smaller discounted transfer distance to provide a better self-supervisory signal. When $p = \infty$, only the maximum element in the discounted transfer vector, namely, the longest part of the total trajectory, needs to be considered, thereby stabilizing the training process. Another potential reason could be that due to the presence of the discount factor, long-term tasks are more heavily impacted by the deviation in the state value calculation, leading to the instability of subgoal planning.

Further observations reveal that when $p = \infty$ and the number of subgoals is two or more, the efficiency of the goal-conditioned policy update is remarkable in the early stages, yet dwindles over time. We conjecture that this may be attributed to the immense challenge of concurrently and randomly sampling more than one subgoal and positioning them on the optimal trajectory. However, it is comparatively simpler when only one subgoal. Based on overall observation, we selected $p = \infty$ and $k = 1$ due to their demonstrable stability and efficiency.

## D ADDITIONAL EXPERIMENTS

### D.1 SUBGOAL LOSS

We calculate the subgoal losses for VE, VE without curriculum learning, and RIS on AntMaze-U (see Figure 10), derived by computing the Mean Squared Error (MSE) in correlation with the oracle subgoal.

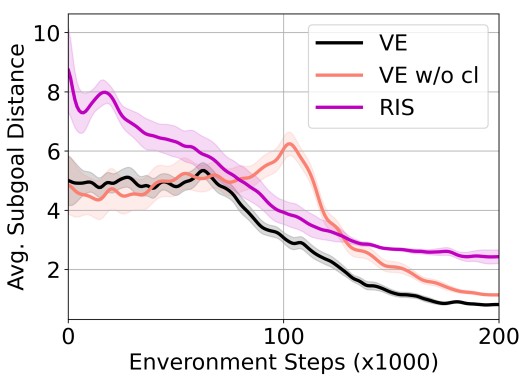

Figure 10: Subgoal losses of VE, VE without curriculum learning (cl) and RIS on AntMaze-U.

## D.2 RESULTS ON THE MOST DIFFICULT TASKS

We evaluate our method on all maps of AntMaze (see Figure 11), comparing the average success rate with RIS on the most difficult tasks (all other methods are 0).

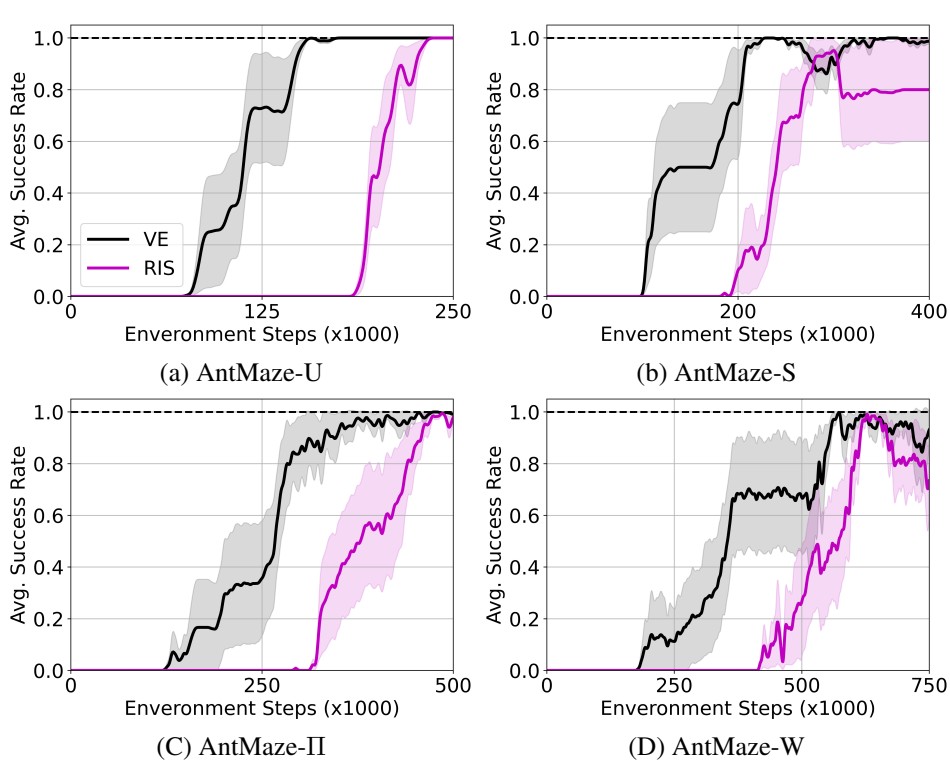

Figure 11: Average success rate of VE and RIS in the most difficult task.

In Figure 12, we also visualize the final states on the map.

## D.3 VISUALIZATION OF SUBGOAL PLANNING

We visualize the subgoal planning of VE in all navigation tasks, and the results are shown in the Figure 13.

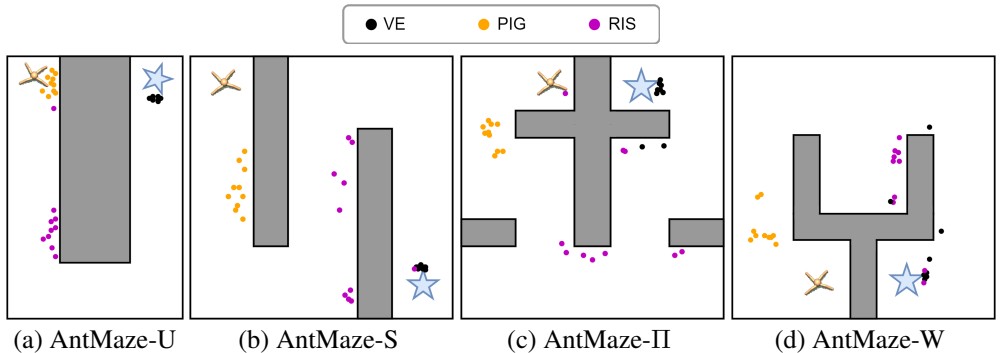

Figure 12: The results of three methods on the most difficult tasks in four navigation environments. We test at different checkpoints: AntMaze-U at 150k environment steps; AntMaze-S at 200k environment steps; AntMaze-Pi at 300k environment steps and AntMaze-W at 500k environment steps.

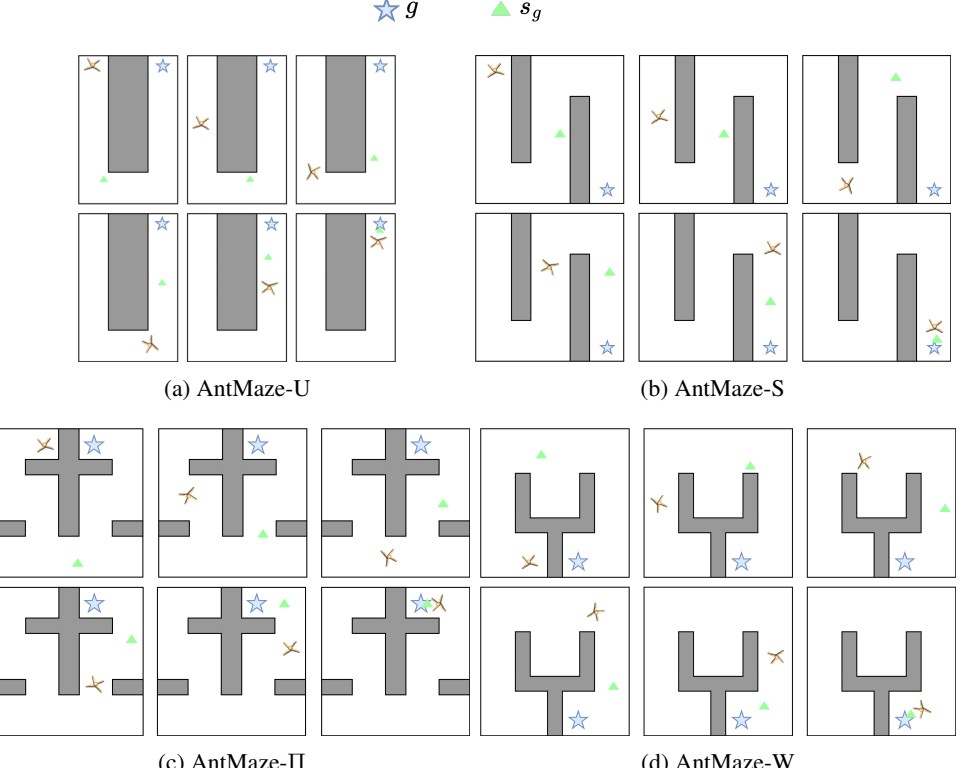

Figure 13: Visualization of subgoal planning on all maps of AntMaze.

## E    ADDITIONAL RELATED WORKS

Determining an appropriate goal for the agent remains an ongoing challenge. (Florensa et al., 2018) define a reward to quantify the agent's proficiency during the testing phase. This reward, informed by the most recent test results, serves as a measure throughout the training process to identify areas where the agent excels or needs improvement. This evaluative information is then used to set goals for the subsequent phase. As the agent's capabilities improve, the goals gradually become more challenging until the desired objectives are achieved. (Hu et al., 2023) introduce PEG, an approach based on a world model that uses the Go-Explore framework to facilitate exploration. To reduce the computational burden associated with evaluating the agent's capabilities, PEG extrapolates the process within the world model instead of directly interacting with the real environment. Skew-Fit (Pong et al., 2019) samples states from a replay buffer and assigns more weight to rare states. It then trains a generative model with these weighted samples. By sampling new states with goals proposed from this generative model, a higher entropy state distribution is obtained in the next iteration. (Nair & Finn, 2019) train a subgoal generative model and a transfer prediction model. For a given goal, a series of subgoals are generated from the subgoal generative model, and the transfer prediction model is used to plan a sequence of actions to achieve these subgoals. In contrast, our work focuses on learning strategies from virtual experiences. Our method can be enhanced by selecting suitable virtual goals through curriculum learning or alternative approaches, and by utilizing improved subgoal planning techniques.

## F    ENVIRONMENT DETAILS

### F.1    REACHER

A robotic arm aims to make its end-effector reach the target position on 3D space. The state space of the arm is 17-dimension, including the positions, angles, and velocities of itself, and the action-space is 7-dimension. Initial point and target goal are set randomly at the start of episode both at training and test time. The agent should reach the target point within 100 steps.

### F.2    SAWYER

It is a vision-based robotic manipulation task where an agent controls a 2 DoF robotic arm from image input and must manipulate a puck positioned on the table. The agent observes a $84 \times 84$ RGB image showing a top-down view of the scene. The dimension of the workspace are 40cm × 20cm and the puck has a radius of 4cm. We consider that the goal is achieved if both the arm and the puck are within 5cm of their respective target positions.

### F.3    ANTMAZE

A quadruped ant robot is trained to reach a random goal from a random location. The states of ant is 31-dimension, including positions and velocities. An ant should reach the target point within 600 steps.

## G    IMPLEMENTATION DETAILS

From Equation 6, we can see that the number of subgoals $k$ controls the distance between them. The high-level policy outputs the distribution of subgoal. For each $k$, we sample 10 subgoals from the distribution to calculate the mean in Equation 9.

The choice of task probability density estimation model mainly considers computational efficiency and support for high-dimensional data (e.g., the state space in AntMaze has 31 dimensions). We compare the results of the kernel density estimation (KDE) (Rosenblatt, 1956) and Flow (Papamakarios et al., 2021), and found that the KDE method only supports low-dimensional samples (e.g., x and y coordinates), so we finally adopt the Flow model based on RealNVP (Dinh et al., 2016) and follow basic parameters. More importantly, when a large number of candidate virtual goals need to be processed, the calculation speed of the Flow model is significantly better than that of the KDE model.

| Hyperparameter | Reacher | AntMaze | Sawyer |
|---|---|---|---|
| Q hidden sizes | [256, 256] | [256, 256] | [256, 256] |
| Policy hidden sizes | [256, 256] | [256, 256] | [256, 256] |
| Subgoal prediction hidden sizes | [256, 256] | [256, 256] | [256, 256] |
| Hidden activation functions | ReLU | ReLU | ReLU |
| Batch size | 1024 | 2048 | 1024 |
| Replay buffer size | 1e6 | 1e6 | 1e5 |
| Discount factor $\gamma$ | 0.99 | 0.99 | 0.99 |
| polyak for target networks | 5e-3 | 5e-3 | 5e-3 |
| Critic learning rate | 1e-3 | 1e-3 | 2e-3 |
| Policy learning rates | 1e-3 | 1e-3 | 5e-3 |
| Subgoal prediction learning rate | 1e-4 | 1e-4 | 1e-3 |
| Flow model learning rate | 1e-3 | 1e-3 | 1e-3 |
| Flow model learning batch size | 1024 | 1024 | 1024 |
| actual goal:virtual goal | 0.5:0.5 | 0.5:0.5 | 0.5:0.5 |
| $\alpha$ | 0.1 | 0.1 | 0.1 |

Table 1: Hyperparameters for VE.

The goal space of the Reacher environment is defined as $(x, y, z)$ in Euclidean space. We transform each state of the agent into a corresponding achieved goal, and use the state-value function $V$ alone to estimate the distance between the achieved goal and the desired goal.

In the Sawyer environment, we use the same encoder as RIS (Chane-Sane et al., 2021). The encoder compresses the received 84 X 84 RGB image into a 32-dimensional vector, and it updates synchronously with the critic network. The gradients of the encoder are subsequently truncated.

## H HYPERPARAMETERS

The hyperparameters we adopt in VE are shown in the Table 1. Each comparison baseline uses the best parameters of the source code. We divide the experimental environment into two classes: state-based "Reacher" and "AntMaze" and pixel-based "Sawyer". In "Reacher" and "AntMaze," we search the learning rate (subgoal prediction, policy, critic) from the candidates 0.0001, 0.001, 0.01 for VE with a $3 \times 3 \times 3$ grid. We use the same parameters in these environments. However, in "Sawyer," we discover that appropriately increasing the learning rate is a better choice. For the baselines (RIS, PIG, HIGL), they use the same experimental environment as ours. So we keep the parameters in the original paper (as the authors have tuned). For the others (GCSL, SAC+HER), we adjust them using a parameter search method similar to VE.

