# OpenReview forum: "Goal-Conditioned Reinforcement Learning with Virtual Experiences"
_ICLR.cc/2025/Conference — Submitted to ICLR 2025_

### Official Review · Reviewer_6ueC · 2024-10-19

**Soundness:** 3
**Presentation:** 2
**Contribution:** 2
**Rating:** 6
**Confidence:** 3

**Summary:**

This paper proposes a novel framework called Virtual Experience (VE) that leverages curriculum learning and subgoal planning to improve sample efficiency and tackle long-horizon tasks. It uses task-probability-density-based curriculum learning and self-supervised subgoal planning. VE integrates low-level skills from diverse experiences to improve exploration efficiency, especially for long-horizon tasks with sparse rewards. The experiments demonstrate VE's superior performance in navigation and robotic control tasks, significantly boosting sample efficiency and handling complex tasks that require multiple goals. The paper also explores the balance between actual and virtual experiences in optimizing learning.

**Strengths:**

* This paper provides extensive experimental validation in a variety of environments.
* The authors offer comprehensive visualizations and a detailed experimental setup.
* This paper extends the HER method with virtual experience from the whole replay buffer, offering a more flexible approach to goal relabeling.

**Weaknesses:**

* Several formula symbols are not clearly presented, which significantly hinders readability and comprehension.
* The paper assumes that virtual goals can be useful without fully addressing the potential for introducing noise or unrealistic transitions. The framework’s reliance on implicit curriculum learning to mitigate this issue may not be effective in more complex, real-world settings where noise cannot be easily filtered out.

**Questions:**

* I do not fully understand how subgoals are selected based on task probability density $e$ for curriculum learning.
* I am confused why introducing only a single subgoal (i.e. $k=1$) in AntMaze yields the best results.
* Can the entire framework be adapted to integrate different lower-level off-policy algorithms, such as HER?

---

> ### Author Response · Authors · 2024-11-21
>
> We thank you for your reviews and address your concerns as follows:
>
> Q1: I do not fully understand how subgoals are selected based on task probability density $e$ for curriculum learning.
>
> A1: The method for training task probability density model described in Appendix A. And It can be viewed in conjunction with Fig. 2 (Left). After training, we randomly sample different states (e.g. 5 in Fig.2) from the replay buffer as virtual goals $g_i$. For the agent's current state $s$, we calculate $e_i(s, g_i)$ and filter appropriate virtual goals through "0.8$\bar e$<$e_i(s,g_i)$<1.2$\bar e$" (line 233), with the expectation that it is located on the boundary.
>
> Q2: I am confused why introducing only a single subgoal (i.e. k=1) in AntMaze yields the best results.
>
> A2: This is a good question. We believe that this conclusion is closely related to our self-supervised learning approach. Since an effective high-level policy improvement relies on sampling appropriate subgoals, when the number of subgoals increases (greater than 1), it becomes exceptionally difficult to obtain subgoals that fall precisely on the optimal path through random sampling.
>
> Q3: Can the entire framework be adapted to integrate different lower-level off-policy algorithms, such as HER?
>
> A3: Our goal relabeling method is indeed an extension of HER, with the key aspect being the introduction of virtual experiences constructed through virtual goals. Observing the results in Fig. 7 (left), when only the HER method is used ("a:v=1.0:0"), VE's performance suffers significantly, performing similarly to "SAC+HER," which indicates that our imitation learning does not fulfill its intended role.
>
> Finally, we want to indicate that the noise introduced by virtual experiences mentioned in the paper is actually policy divergence. Specifically, there is a divergence between the old policy $\pi(a^i_t|s^i_t,g^i)$ stored in the replay buffer and the new policy $\pi(a^i_t|s^i_t,g^j)$ after relabeling. The divergence caused by HER's relabeling method is not significant, thanks to its reliance on real trajectories. However, virtual goals lead to more severe divergence. Therefore, what VE imitates are the subgoal-conditioned policies $\pi^{prior}(s,s_g)$, rather than the reconstructed original actions $a^i_t$ (which include noise in Eq. (5)) in the virtual experiences.

---

> > ### Comment · Reviewer_6ueC · 2024-11-22
> >
> > Thank you for your response. Due to some lack of clarity in paper writing, I raised a few questions about the article that I didn't fully understand. Your reply has addressed most of my concerns, and I will increase the score accordingly.

---

> ### Author Response · Authors · 2024-11-22
>
> We thank you for your recognition of this paper.

---

### Official Review · Reviewer_SKRR · 2024-11-02

**Soundness:** 3
**Presentation:** 2
**Contribution:** 2
**Rating:** 5
**Confidence:** 3

**Summary:**

Goal-conditioned environments are MDPs where the policy reward aligns with reaching a given end goal. The sample complexity of such problems can be significantly reduced with augmentation techniques such as hindsight experience replay (HER), where visited states in a trajectory are relabeled as goals and used to augment the policy training procedure. This work suggests an expanded augmentation, that broadly aims to break the goal-based task to several sub-goals using a higher level policy, and learn reaching the sub-goals using a lower level policy. The high-level policy can be trained directly with the value function of the lower level policy.

**Strengths:**

* The idea of a high-level policy learning the sub-goals directly is elegant.
* The evaluations show a wide gap between the proposed approach and baselines.

**Weaknesses:**

I'm fond of this work, but there are several things I found concerning. An explanation by the authors could potentially be helpful.

* While §4.2 and §4.3 were well-written, I found §4.1 very hard to grasp.

    What is $j$, and why is it that if $i=j$, Eq. 5 is equivalent to Eq. 4? What is the "task probability density" trying to learn? What does the condition (e.g., the range filter) mean from a high-level perspective? Why would there be a "lack of connection between state transitions", as described in line 238? It doesn't seem like state transitions are being tinkered with in Eq. 5.

    Overall, I would suggest a major rewrite of this section.
* How was VE tuned for the experiments in §5.2? It seems like VE was tuned per each environment separately, but how was this done without tuning on the "test" problem (the goal conditioned environment that you evaluate on)?

Some minor nitpicks:
* Several "Swayer" typos: Figure 3, F.2, etc.
* There is no $j$ in Eq. 5.

**Questions:**

* How was VE tuned? This is related to the weaknesses comment I made.
* What is the main difference between this work and other works that employ sub-goals? Is it that other works attain sub-goals without learning (e.g. the midpoint, bisection or graph-based methods)?
* Are there any ablation experiments that show the effectiveness of Eq. 9 in §4.3? If I understood correctly, Eq. 9 played the role of a soft regularization over the policy. It would be interesting to show how important it is, but I could not find it in §5.3.
* I found it difficult to get a final takeaway from Figure 5. Did I miss something? Is there any information we can deduce from it?
* In line 317, it is stated that "we use a entire 31-dimensional state space as the goal space". Why? Without justification, this comes off as an arbitrary decision. Even if the reason is trivial, it would be helpful to mention it somewhere, or link to an explanation in the appendix.
* What is the concrete definition of "Uniform smoothing" in line 357? Either a citation is needed, or a reasonable technical explanation. Otherwise it is difficult to know what artifacts might have been caused by the smoothing.

---

> ### Author Response · Authors · 2024-11-21
>
> We thank you for your reviews and address your concerns as follows:
>
> Q1: What is j, and why is it that if i=j, Eq. 5 is equivalent to Eq. 4? What is the "task probability density" trying to learn? What does the condition (e.g., the range filter) mean from a high-level perspective? Why would there be a "lack of connection between state transitions", as described in line 238? It doesn't seem like state transitions are being tinkered with in Eq. 5.
>
> A1: We apologize for any confusion caused by our previous statement. We hope this explanation clarifies the issue. In Eq. (5), we employ a more general relabeled goal notation (bold) $g$, to signify that our defined generalized goal re-labeling can incorporate actual, virtual, or custom-made goals (Eqs. (4) and (5) mix $s$ and $g$, as we assume $S=G$). Superscripts $i$ and $j$ are used to distinguish whether the relabeled goal comes from the same trajectory as $s^i_t$. Since HER only considers goals from the same trajectory, $i=j$. However, we also use virtual goals from different trajectories to construct the experience needed for learning ($i \neq j$).
>
> As shown in Fig. 2 (Left), we distinguish which virtual goals are at the boundary by learning a task probability density model. This helps us build a step-by-step curriculum (see Fig. 1(b)). The task probability density model does not participate in the optimization process of the high-level policy; it is a relatively independent module used solely for constructing training data.
>
> The term mentioned on line 238 "lack of connection between state transitions" refers to the fact that in GCRL, the policy $\pi(s,g)$ simultaneously considers the current state $s$ and the goal $g$. However, in Eq. (5), both the goal and the reward have been changed (we replace them with actual and virtual goals), weakening (or disrupting) the connection between the new (bold) $g$ and $(s^i_t, a^i_t, s^i_{t+1})$.
>
> Q2: How was VE tuned for the experiments in §5.2? It seems like VE was tuned per each environment separately, but how was this done without tuning on the "test" problem (the goal conditioned environment that you evaluate on)?
>
> A2: The proposed VE contains several core hyperparameters, which we have evaluated in detail and kept consistent during training and testing. In Appendix Fig. 9, we assess the impact of the $p$-norm and the number of subgoals $k$ on the results, and we use $p=\infty $ and $k=1$ in all experiments. For the screening by task probability models, we have set the range to (0.8, 1.2). The proportion of actual goals to virtual goals is evaluated in Fig. 7 (left), and we consistently use "a:v=0.5:0.5" in all experiments. In summary, the hyperparameters used by VE are generally consistent across all environments, and there is no difference between training and testing. Notably, curriculum learning and subgoal planning are active only during training, whereas testing relies solely on trained goal-conditioned policiy interacting with the environment.
>
> Q3: What is the main difference between this work and other works that employ sub-goals? Is it that other works attain sub-goals without learning (e.g. the midpoint, bisection or graph-based methods)?
>
> A3: "Midpoint":[1] This study focuses on offline RL, which differs from the online RL discussed in this paper. It utilizes Transformer to predict intermediate subgoals. "Bisection":[2] Connects subgoal planning with the dynamic programming equation for the *all pairs shortest path* (APSP) problem, which naturally advantages navigation and obstacle avoidance tasks but is challenging to extend to high-dimensional goal spaces (VE uses the complete state space as the goal space). "Graph-based":[3,4] is the main comparative method in this paper. We believe that a graph structure is an effective form of preserving historical knowledge, but it suffers from low training efficiency and suboptimality. Therefore, we provide knowledge from different tasks by constructing virtual experiences and learn from them via VE. The biggest difference between VE and the methods mentioned above is its use of self-supervised learning for subgoal planning, and the knowledge from historical experiences can be seamlessly integrated into general reinforcement learning approaches via a straightforward self-imitation process. In our understanding, the above methods all rely on learning to master how to plan subgoals.

---

> > ### Author Response · Authors · 2024-11-21
> >
> > Q4: Are there any ablation experiments that show the effectiveness of Eq. 9 in §4.3? If I understood correctly, Eq. 9 played the role of a soft regularization over the policy. It would be interesting to show how important it is, but I could not find it in §5.3.
> >
> > A4: Your understanding is correct; we can regard the imitation process as a regularization term. Since all the design elements of VE ultimately focus on the imitation learning part in Eq.(9), we consider "SAC+HER" as the baseline without imitation learning. It is important to note that "SAC+HER" does not use virtual experiences, as this would severely compromise its stability.
> >
> > Q5: I found it difficult to get a final takeaway from Figure 5. Did I miss something? Is there any information we can deduce from it?
> >
> > A5: For Fig. 5, we recommend analyzing it in conjunction with Fig. 1. In Fig. 1(a), we point out that using state probability density for estimation in existing methods is not effective in assessing the reachability of tasks (s, g). However, using task probability density estimation can help us construct a step-by-step curriculum (Figure 1(b)), with the specific visualization results shown in Fig. 5.
> >
> > Q6: In line 317, it is stated that "we use a entire 31-dimensional state space as the goal space". Why? Without justification, this comes off as an arbitrary decision. Even if the reason is trivial, it would be helpful to mention it somewhere, or link to an explanation in the appendix.
> >
> > A6: Thank you for raising the relevant concerns. Using the full state space as the goal space is one of the advantages in this paper. In contrast, previous work [3,4] adopted a simplified goal space, which greatly reduced the difficulty of subgoal planning. However, VE is able to capture task-related information from the complete state space (e.g. in navigation tasks, the (x, y) coordinates of the subgoals are accurate).
> >
> > Q7: What is the concrete definition of "Uniform smoothing" in line 357? Either a citation is needed, or a reasonable technical explanation. Otherwise it is difficult to know what artifacts might have been caused by the smoothing.
> >
> > A7: We agree with the writing suggestions you provided and have added the references accordingly [4]. You can read our latest revised version.
> >
> >
> >
> >
> >
> > [1] Badrinath, A., Flet-Berliac, Y., Nie, A., & Brunskill, E. Waypoint transformer: Reinforcement learning via supervised learning with intermediate targets. (NeurIPS2024)
> >
> > [2] Jurgenson, T., Avner, O., Groshev, E., & Tamar, A. Sub-goal trees a framework for goal-based reinforcement learning. (ICML2020)
> >
> > [3] Kim, J., Seo, Y., & Shin, J. Landmark-guided subgoal generation in hierarchical reinforcement learning. (NeurIPS2021)
> >
> > [4] Kim, J., Seo, Y., Ahn, S., Son, K., & Shin, J. Imitating Graph-Based Planning with Goal-Conditioned Policies. (ICLR2023)

---

> > ### Comment · Reviewer_SKRR · 2024-11-25
> >
> > I thank the authors for their responses.
> >
> > ---
> >
> > > Q1: What is j, and why is it that if i=j, Eq. 5 is equivalent to Eq. 4? What is the "task probability density" trying to learn? What does the condition (e.g., the range filter) mean from a high-level perspective? Why would there be a "lack of connection between state transitions", as described in line 238? It doesn't seem like state transitions are being tinkered with in Eq. 5.
> > >
> > > A1: ...
> >
> > Thank you for the clarification. Could the authors also clarify the concept of the range filter?
> >
> > > The term mentioned on line 238 "lack of connection between state transitions" refers to the fact that in GCRL, the policy $\pi(s, g)$ simultaneously considers the current state $s$ and the goal $g$. However, in Eq. (5), both the goal and the reward have been changed (we replace them with actual and virtual goals), weakening (or disrupting) the connection between the new (bold) $g$ and $(s^i_t, a^i_t, s^i_{t+1})$
> >
> > State transitions are the change in the MDP state, i.e., a probability distribution over the next state defined by a transition kernel $\mathcal{T}(s_{t+1}|s_t, a_t)$. If you are only changing the goals for states $s_t$ and $s_{t+1}$, you are not disrupting state transitions. While these "synthetic" states with virtual goals are not real, they do conform to the MDP dynamics; the goal, at least in most goal-oriented MDPs I know, does not affect the transition kernel. The goal affects the policy. Perhaps I am missing a key detail here?
> >
> > ---
> >
> > > Q2: How was VE tuned for the experiments in §5.2? It seems like VE was tuned per each environment separately, but how was this done without tuning on the "test" problem (the goal conditioned environment that you evaluate on)?
> > >
> > > A2: ...
> >
> > When I asked about hyperparameter tuning, I was referring to Table 1 in the Appendix. The learning rates of the networks (subgoal prediction, policy, critic) are different across 3 environments, and are not consistent. How were the hyperparameters in Table 1 tuned? I could not find hyperparameters for baselines. The same question can be posed for your experiments with these baselines. Did you tune the baselines, and if so, how?
> >
> > ---
> >
> > > Q3: What is the main difference between this work and other works that employ sub-goals? Is it that other works attain sub-goals without learning (e.g. the midpoint, bisection or graph-based methods)?
> > >
> > > A3: ... The biggest difference between VE and the methods mentioned above is its use of self-supervised learning for subgoal planning, and the knowledge from historical experiences can be seamlessly integrated into general reinforcement learning approaches via a straightforward self-imitation process. In our understanding, the above methods all rely on learning to master how to plan subgoals.
> >
> > Doesn't VE also rely on learning to master how to plan subgoals (via the high-level subgoal policy)?
> >
> > I do agree that the main point of difference is the use of self-supervised learning/reinforcement learning. Discussing this (and above points) is necessary in the paper, somewhere in either the introduction section or §4.1 where you first talk about subgoals in VE.
> >
> > I would suggest rewriting line 52/61, and instead of contrasting VE to HER, explain prior subgoal-based HER approaches and contrast VE to them. Then it becomes clearer where your proposition lies compared to prior work.
> >
> > ---
> >
> > > Q5: I found it difficult to get a final takeaway from Figure 5. Did I miss something? Is there any information we can deduce from it?
> > >
> > > A5: ...
> >
> > I understand. But when I read the paper, the pattern in the density maps in Figure 5 were too subtle to warrant space in the main text. The state density maps do not provide information at all. I would suggest moving this figure to the appendix, and using the saved space to (1) better explain VE in §4 and (2) its relation to subgoal-based approaches.
> >
> > ---
> >
> > > Q6: In line 317, it is stated that "we use a entire 31-dimensional state space as the goal space". Why? Without justification, this comes off as an arbitrary decision. Even if the reason is trivial, it would be helpful to mention it somewhere, or link to an explanation in the appendix.
> > >
> > > A6: ...
> >
> > I am confused. Isn't the goal always a point in some N-dimensional space? Is the goal space 31 dimensional? The environments in Figure 3 seem to be 3 dimensional at most. Perhaps more elaboration would help here.
> >
> > ---
> >
> > > Q7: What is the concrete definition of "Uniform smoothing" in line 357? Either a citation is needed, or a reasonable technical explanation. Otherwise it is difficult to know what artifacts might have been caused by the smoothing.
> > >
> > > A7: We agree with the writing suggestions you provided and have added the references accordingly [4]. You can read our latest revised version.
> >
> > I just checked and line 357 seems to not include a reference for uniform smoothing. How is [4] related?

---

> ### Author Response · Authors · 2024-11-25
>
> Dear Reviewer SKRR
>
> Thanks for your time and comments on our work!
>
> We have tried our best to address the concerns and provided detailed responses to all your comments and questions.
>
> Would you mind checking our response and confirming whether you have any further questions?
>
> Best regards, Authors of #5840

---

> ### Author Response · Authors · 2024-11-27
>
> We thank you for your response and constructive comments.
>
> > Could the authors also clarify the concept of the range filter?
>
> Sorry for the misunderstanding. We want to clarify that "range" and "filter" should be understood separately instead of being combined. The "filter" means that we apply the "task probability density" $e$ in curriculum planning, acting as a "filter" to screen suitable virtual goals $g$ from random sampling. While the "range" means a simple "range" constraint (0.8$\bar e$<$e(s,g)$<1.2$\bar e$) for the filtering method that is adopted in the proposed method.
>
> > State transitions are the change in the MDP state, i.e., a probability distribution over the next state defined by a transition kernel $T(s_{t+1}|s_t,a_t)$. If you are only changing the goals for states $s_t$ and $s_{t+1}$, you are not disrupting state transitions. While these "synthetic" states with virtual goals are not real, they do conform to the MDP dynamics; the goal, at least in most goal-oriented MDPs I know, does not affect the transition kernel. The goal affects the policy. Perhaps I am missing a key detail here?
>
> We agree with your understanding of state transitions in the MDP, even under goal-conditioned. We want to clarify that “lack of connection” discusses the problem between the original state transition and the relabeled goal, and it does not represent that whether the state transition itself has been disrupted. So we modify as: "Firstly, from a policy perspective, there is a lack of connection between the original state transitions $(s^i_t, a^i_t, s^i_{t+1})$  and the relabeled goals $g$. This reduces the accuracy of policy evaluation."
>
> The action $a^i_t$ in the original experience is a decision made by the policy $\pi$ under the conditions of the state $s^i_t$ and the original goal $g^i$, so the state transition caused by the action is strongly correlated with the original goal, while the relabeled goal $g$ in the reconstructed data has lost its relevance to the original state transition  $(s^i_t, a^i_t, s^i_{t+1})$  and cannot reflect a correct policy, thus reducing the accuracy of policy evaluation.
>
> > When I asked about hyperparameter tuning, I was referring to Table 1 in the Appendix. The learning rates of the networks (subgoal prediction, policy, critic) are different across 3 environments, and are not consistent. How were the hyperparameters in Table 1 tuned? I could not find hyperparameters for baselines. The same question can be posed for your experiments with these baselines. Did you tune the baselines, and if so, how?
>
> Thanks for your comments. Our comparisons are fair. We divide the experimental environment into two classes: state-based "Reacher" and "AntMaze" and pixel-based "Sawyer". In "Reacher" and "AntMaze," we search the learning rate (subgoal prediction, policy, critic) from the candidates {0.0001, 0.001, 0.01} for VE with a $3\times 3 \times 3$ grid. We use the same parameters in these environments. However, in "Sawyer," we discover that appropriately increasing the learning rate is a better choice. For the baselines (RIS, PIG, HIGL), they use the same experimental environment as ours. So we keep the parameters in the original paper (as the authors have tuned). For the others (GCSL, SAC+HER), we adjust them using a parameter search method similar to VE.
>
> > Doesn't VE also rely on learning to master how to plan subgoals (via the high-level subgoal policy)?
> >
> > I do agree that the main point of difference is the use of self-supervised learning/reinforcement learning. Discussing this (and above points) is necessary in the paper, somewhere in either the introduction section or §4.1 where you first talk about subgoals in VE.
> >
> > I would suggest rewriting line 52/61, and instead of contrasting VE to HER, explain prior subgoal-based HER approaches and contrast VE to them. Then it becomes clearer where your proposition lies compared to prior work.
>
> Thanks for your good comment. The methods introduced in the paper can be classified into into learning-based and search-based methods. Learning-based methods generally require only a single forward pass through the network for subgoal planning, making them more efficient and user-friendly than graph search-based methods. In this paper, both VE and RIS employ a learning-based approach, whereas PIG and HIGL utilize graph search techniques. Other reviewed methods do not incorporate subgoal planning. Thanks for your good suggestions, we have made corresponding revisions in the paper.
>
> > I understand. But when I read the paper, the pattern in the density maps in Figure 5 were too subtle to warrant space in the main text. The state density maps do not provide information at all. I would suggest moving this figure to the appendix, and using the saved space to (1) better explain VE in §4 and (2) its relation to subgoal-based approaches.
>
> Thanks for your good suggestion. We have revised Sec. 4 in the reversion.

---

> > ### Author Response · Authors · 2024-11-27
> >
> > > I am confused. Isn't the goal always a point in some N-dimensional space? Is the goal space 31 dimensional? The environments in Figure 3 seem to be 3 dimensional at most. Perhaps more elaboration would help here.
> >
> > We apologize for the confusion. The goal is usually defined as a point in N-dimensional space. Take "AntMaze" as an example, the state space is 31-dimensional, including the x and y coordinates of the ant. We do not deny that the navigation task is only related to the x and y coordinates, so some works usually set the goal to 2 dimensions. Different from this, we follow the setting of RIS [1] and set the goal space to 31 dimensions (same with full state space).
> >
> > When generating random tasks, we only consider key features (the 2D coordinates of the ant) and pad non-critical feature dimensions with 0 to maintain full dimensionality (31 dimensions ). The difference is that the relabeled goal is derived from the state, where the value of non-critical features is not 0. This requires high-level policy and value functions to identify the state features that are truly relevant to the task from the reward signal. This means that the x and y dimensions of the predicted subgoals (31 dimensions) are accurate. Similarly, when calculating the value, the Critic is more concerned about the x and y dimensions. However, specifying a 2D goal space simplifies this process. Although this setting slightly increases the learning difficulty, VE is equally effective in the full 31-dimensional goal space as well as in the 2-dimensional goal space.
> >
> > > I just checked and line 357 seems to not include a reference for uniform smoothing. How is [4] related?
> >
> > We apologize for the confusion caused by the imprecise representation "Uniform smoothing". Our intention is to emphasize that the smoothing of the curves is equal. To avoid ambiguity, we use a similar expression as in [4]: "For visual clarity, we smooth all the curves equally."
> >
> >
> >
> > [1] Chane-Sane, E., Schmid, C., & Laptev, I. Goal-conditioned reinforcement learning with imagined subgoals. (ICML2021)
> >
> > [4] Kim, J., Seo, Y., Ahn, S., Son, K., & Shin, J. Imitating Graph-Based Planning with Goal-Conditioned Policies. (ICLR2023)

---

> > ### Comment · Reviewer_SKRR · 2024-11-27
> >
> > I thank the authors for their responses. The changes in the paper are welcome; section 4 reads with more clarity but I strongly recommend further revisions to this section. For instance, a discussion on how the policy updates change when using virtual goals (Eq. 5) is useful; In HER it is very clear how the hindsight trajectory can be recreated but with a virtual goal this needs further explanation. This explanation can also clarify the learning difficulties introduced with virtual goals, and why sub-goal planning then becomes useful.
> >
> > ---
> >
> > > Thanks for your comments. Our comparisons are fair. We divide the experimental environment into two classes: state-based "Reacher" and "AntMaze" and pixel-based "Sawyer". In "Reacher" and "AntMaze," we search the learning rate (subgoal prediction, policy, critic) from the candidates {0.0001, 0.001, 0.01} for VE with a  grid. We use the same parameters in these environments. However, in "Sawyer," we discover that appropriately increasing the learning rate is a better choice. For the baselines (RIS, PIG, HIGL), they use the same experimental environment as ours. So we keep the parameters in the original paper (as the authors have tuned). For the others (GCSL, SAC+HER), we adjust them using a parameter search method similar to VE.
> >
> > I saw the changes in the paper. While appreciated, for reproducibility it is best to report hyperparameters for all schemes, including baselines.
> >
> > ---
> >
> > > We apologize for the confusion caused by the imprecise representation "Uniform smoothing". Our intention is to emphasize that the smoothing of the curves is equal. To avoid ambiguity, we use a similar expression as in [4]: "For visual clarity, we smooth all the curves equally."
> >
> > Perhaps I have miscommunicated my point. You have to define what "smoothing" means. Is it an EWMA smoothing? It is smoothing along multiple seeds? Is it a kernel based smoothing? It is a low-pass filter?
> >
> > Reference [4] is not the appropriate citation; you need to cite the original smoothing method or library. Works that have used similar "smoothing" techniques are second-hand citations; plus, reference [4] also doesn't mention what the "smoothing" is formally.
> >
> > ---
> >
> > > We apologize for the confusion. The goal is usually defined as a point in N-dimensional space. Take "AntMaze" as an example, the state space is 31-dimensional, including the x and y coordinates of the ant. We do not deny that the navigation task is only related to the x and y coordinates, so some works usually set the goal to 2 dimensions. Different from this, we follow the setting of RIS [1] and set the goal space to 31 dimensions (same with full state space).
> >
> > I'm interested in why such an evaluation setting makes sense. Why follow the setting of RIS [1], and not other prior work that use the 2-dimensional goal?
> >
> > Again, the purpose of this clarification is for this decision to not seem arbitrary. A logical justification is what is missing. Are there situations where using a 31-dimensional state as a proxy for the goal makes more sense than directly using the 2-dimensional goal?

---

> > > ### Author Response · Authors · 2024-11-29
> > >
> > > Thanks for your continued attention and review of this paper.
> > >
> > > > The changes in the paper are welcome; section 4 reads with more clarity but I strongly recommend further revisions to this section. For instance, a discussion on how the policy updates change when using virtual goals (Eq. 5) is useful; In HER it is very clear how the hindsight trajectory can be recreated but with a virtual goal this needs further explanation. This explanation can also clarify the learning difficulties introduced with virtual goals, and why sub-goal planning then becomes useful.
> > >
> > > Thanks for your constructive feedback, we will add this to Sect. 4 in the camera ready version.
> > >
> > > > I saw the changes in the paper. While appreciated, for reproducibility it is best to report hyperparameters for all schemes, including baselines.
> > >
> > > Thanks for your recognition of the rationality of the experimental parameters in this paper. We will supplement the main hyperparameters used by the baseline methods.
> > >
> > > > Perhaps I have miscommunicated my point. You have to define what "smoothing" means. Is it an EWMA smoothing? It is smoothing along multiple seeds? Is it a kernel based smoothing? It is a low-pass filter?
> > > >
> > > > Reference [4] is not the appropriate citation; you need to cite the original smoothing method or library. Works that have used similar "smoothing" techniques are second-hand citations; plus, reference [4] also doesn't mention what the "smoothing" is formally.
> > >
> > > We apologize for misunderstanding your question. Specifically, we calculate the mean and standard deviation of the results for the four random seeds. The solid line represents the mean and the shaded area represents the standard deviation. For visual clarity, we apply Gaussian filtering with the same parameters to all curves in each experiment to ensure fairness.
> > >
> > > > I'm interested in why such an evaluation setting makes sense. Why follow the setting of RIS [1], and not other prior work that use the 2-dimensional goal?
> > > >
> > > > Again, the purpose of this clarification is for this decision to not seem arbitrary. A logical justification is what is missing. Are there situations where using a 31-dimensional state as a proxy for the goal makes more sense than directly using the 2-dimensional goal?
> > >
> > > To better explain this problem, let's assume a task in which a robotic arm manipulates an object. From the perspective of the goal, we only care whether the object is placed in the specified location, but when simplifying the task through subgoals, we will additionally consider the posture and speed of the robotic arm as part of the subgoal, not just the position of the object. Although in this example, we can manually define each dimension of the goal and subgoal. But in more general scenarios (including pixel-based tasks), we hope that the agent can learn features that are truly related to the task reward from the complete goal (state) space.
> > >
> > > In the test environment of this paper, RIS is the only method other than VE that can adapt to both state-based tasks and pixel-based tasks. We think this setting of not simplifying the goal and subgoal space is more universal. For state-based tasks, each dimension is derived from sensors and is a feature with accurate semantic information, so it is easy for us to filter features. However, pixel-based tasks contain a lot of background noise, and it is obviously difficult to remove them. We believe that a more general goal/subgoal prediction method should be able to identify features related to reward signals from a feature space containing noise.

---

> > > > ### Comment · Reviewer_SKRR · 2024-12-01
> > > >
> > > > > To better explain this problem, let's assume a task in which a robotic arm manipulates an object. From the perspective of the goal, we only care whether the object is placed in the specified location, but when simplifying the task through subgoals, we will additionally consider the posture and speed of the robotic arm as part of the subgoal, not just the position of the object. Although in this example, we can manually define each dimension of the goal and subgoal. But in more general scenarios (including pixel-based tasks), we hope that the agent can learn features that are truly related to the task reward from the complete goal (state) space.
> > > > >
> > > > > In the test environment of this paper, RIS is the only method other than VE that can adapt to both state-based tasks and pixel-based tasks. We think this setting of not simplifying the goal and subgoal space is more universal. For state-based tasks, each dimension is derived from sensors and is a feature with accurate semantic information, so it is easy for us to filter features. However, pixel-based tasks contain a lot of background noise, and it is obviously difficult to remove them. We believe that a more general goal/subgoal prediction method should be able to identify features related to reward signals from a feature space containing noise.
> > > >
> > > > So the goal in these pixel-based goal-conditioned MDPs is the entire state; this means the typical evaluation setting for them is when subgoals are the same dimension as goals. So why would it make sense to not do the same on the current environments in the paper?
> > > >
> > > > If you believe VE has some superiority in high-dimensional subgoals, why not evaluate on the mentioned pixel-based goal-conditioned MDPs?
> > > >
> > > > It actually would have been better to evaluate on both 2D/3D subgoals, and 31D subgoals for the same environment to highlight that VE solves both, but (likely) less efficiently with 31D subgoals due to having to learn what input features matter most for the return signal.
> > > >
> > > > I am not convinced that 31D subgoals is a well-justified evaluation setting. I understand that the authors have theories about why it might make sense, but there are direct ways to test such theories.
> > > >
> > > > ---
> > > >
> > > > After re-reading the discussion so far, and other comments, I have decided to keep my score. I do not want to discourage the authors; this paper, from a technical standpoint, sounds reasonable. There are some nice ideas to explore and share with the community.
> > > >
> > > > However, the paper needs work on its exposition.
> > > > * Section 4 is vital to explain how VE works, why it works, when will it break, and why is each component necessary. I have read the author revisions, and while helpful, I still believe this section needs a major overhaul. The technical aspects of VE deserve clearer explanations that connect and flow with each other.
> > > >   * The explanation in 4.1 needs to be more formal (e.g., formally define $e(s, g)$, explain how sampling from it works, explain what $\bar{e}$ is, explain how rewards are calculated when adding virtual goals, why virtual goals cause issues). There are some explanations for the provided points, e.g. "there is a lack of connection between the original state transitions" at line 235. But these were not enough. For instance, the sentence in 235 bears no information; what does a lack of connection mean mathematically? Why does it reduce the accuracy of policy evaluations? I know what the intention of these sentences are, and I'm certain the authors know the answers to these questions as well, but their knowledge is not reflected in the text. Your audience cannot (and should not) connect these gaps for you.
> > > >    * For section 4.2, you have to connect the subgoal-planner policy to $e(s, g)$. The change of notations throws the reader off. Why is AWR needed here, compared to more traditional methods such as A2C? Isn't eq. 8 normal policy gradient (no AWR)? What is the optimal sub-goal planner, and how does it interplay with the subgoal-solver policy? How is this idea related to RL2[1]?
> > > >    * For section 4.3, what is $\pi^{\text{prior}}$? What is $\pi_{\theta_m}$? What is the goal of Eq. 9?
> > > > * The evaluations need some reworking; sometimes the evaluation setting is odd (choice of subgoal dimensionality), other times ablations are missing or not explained well, and finally reproducibility information is lacking (baseline parameters, tuning details, clear definitions and references for "smoothing").
> > > >
> > > >
> > > > [1] Duan, Y., Schulman, J., Chen, X., Bartlett, P. L., Sutskever, I., & Abbeel, P. (2016). Rl $^ 2$: Fast reinforcement learning via slow reinforcement learning. arXiv preprint arXiv:1611.02779.

---

### Official Review · Reviewer_TXca · 2024-11-04

**Soundness:** 3
**Presentation:** 3
**Contribution:** 3
**Rating:** 6
**Confidence:** 2

**Summary:**

This paper studies goal-conditioned reinforcement learning, and proposes an approach that 1) learns with a curriculum of relabelled goals, and 2) distills learned behavior for reaching s-> subgoal to learn a model that can reach s -> goal. Across a number of state-based environments, the authors find that the proposed approach significantly outperforms past hierarchical approaches and other GC methods.

**Strengths:**

The goal-conditioned learning problem setting is an important one in reinforcement learning, and this paper offers a nice way of reducing its complexity. I particularly found the idea of distilling policies from a subgoal to a further goal to be a nice one, especially that it helps propagate information that can cleanly be learned at small scales to help shape the noisier long-horizon learning problem.


The results are very thorough: the benefit seems relatively compelling on the studied domains, and the different components of the algorithm are very well ablated. the two different components of the algorithm are well-ablated.

I found the use of visualizations throughout the paper to be useful to help understand the proposed method in the paper -- the visualizations in Figure 5 were particularly useful.

**Weaknesses:**

The importance of the curriculum is not clear to me: for one, it introduces much additional complexity (e.g. the need to train a Flow or RealNVP model, whose details are not well discussed in the paper), it provides seemingly little benefit (in Figure 7, middle), and requires special hyperparameters (e.g. (0.8, 1.2) for state density clipping).

The paper only studies relatively simple state-based domains with intrinsic 2d structure, and most of the analysis is on Antmaze (which has a clear 2d structure, even if this is not the axis being measured) -- this is particularly worrisome, since the high-level policy learning problem gets more complicated in higher dimensional state spaces, and it is unclear how this method generalizes. The paper would be stronger if it had a more complex state-based environment  (e.g. one may consider Franka Kitchen or CALVIN)

Section 4.2 is difficult to follow: it introduces additional notation for the sub-goal sequence that is not explained clearly, and it is not clear whether the mechanism being used is an A2C-like (A2C updates using policy gradient estimates) or if it is AWR-like (which seems to be the form of Equation 8).

Nit: The paper begins the methods section (Section 4) describing how the framework differs from Kim et al, 2023 -- the paper to this point has no description of Kim et al, nor is it made clear prior to this point that the method is building on top of Kim et al, 2023. Especially for readers who are unfamiliar with this work, it would be useful to describe the paper in greater detail in related work, or perhaps softly introduce the ideas in Kim et al otherwise in the paper.


Minor:

Would be good to have non-abbreviated captions in the Figures (e.g. in Figure 7)
The paper has several typos throughout:

Misuse of \cite or \citet instead of \citep throughout the paper
Figure 4 -> Enveronment -> Environment
L893: Swayer -> Sawyer

**Questions:**

1. Why does selecting all virtual goals cause the method to fail? (in Fig 7)
2. It is common in hindsight GC implementations to (with some proportion) sample a goal at random from the replay buffer. This alleviates a lot of the challenges with otherwise only choosing goals on the same trajectory. Is this being done for the baselines (specifically for SAC+HER?)
3. In Equation 9, it seems that $k$ is being overloaded both as an iteration index (e.g. theta_{k+1}), and also as the subgoal index?

---

> ### Author Response · Authors · 2024-11-21
>
> We thank you for your reviews and address your concerns as follows:
>
> Q1: The importance of the curriculum is not clear to me: for one, it introduces much additional complexity (e.g. the need to train a Flow or RealNVP model, whose details are not well discussed in the paper), it provides seemingly little benefit (in Figure 7, middle), and requires special hyperparameters (e.g. (0.8, 1.2) for state density clipping).
>
> A1: Curriculum learning has been shown in certain works to enhance the stability of the learning process and the ultimate success rate [1-3]. In this paper, we regard curriculum learning as a relatively independent module within our proposed framework, which can achieve more significant improvements through better design. Our main contribution is the verification of the advantage of the task probability density model in constructing curricula compared to the state probability density model. Although existing methods still require setting hyperparameters, more elegant automated curriculum methods can be developed in the future. As shown in Fig. 7, the interaction steps at which the agent reaches performance peak are advanced by approximately 29% (190k -> 135k) with the help of curriculum learning, and the standard deviation fluctuations displayed during the training process are smaller, making the process more robust, which also confirms the benefits commonly mentioned in previous works.
>
> Q2: The paper only studies relatively simple state-based domains with intrinsic 2d structure, and most of the analysis is on Antmaze (which has a clear 2d structure, even if this is not the axis being measured) -- this is particularly worrisome, since the high-level policy learning problem gets more complicated in higher dimensional state spaces, and it is unclear how this method generalizes. The paper would be stronger if it had a more complex state-based environment (e.g. one may consider Franka Kitchen or CALVIN)
>
> A2: Thank you for your valuable suggestions. We select the AntMaze environment for analysis, following established research practices [1,4,5], because it offers clear and comprehensible visualization. It is important to note that, in the navigation task, only the two dimensions (x, y) within the complete 31-dimensional state space of AntMaze are directly relevant to the goal. Our proposed subgoal planning method successfully captures this key information without artificially reducing the dimensionality of the goal space (maintaining all 31 dimensions). When extended to more complex state spaces, we believe that the core states remain the 3-dimensional coordinates (x, y, z) that describe the position of the robotic arm or object. We consider that our proposed method also adapts well to this complexity. Furthermore, we conduct experiments in the Sawyer environment, which also features a high-dimensional state space (pixel-based 84$\times$84 RGB), and it demonstrates the potential for planning subgoals based on pixel input. We acknowledge your concerns and incorporate scenarios utilizing pixel input as one of our future research directions.
>
> Q3: Section 4.2 is difficult to follow: it introduces additional notation for the sub-goal sequence that is not explained clearly, and it is not clear whether the mechanism being used is an A2C-like (A2C updates using policy gradient estimates) or if it is AWR-like (which seems to be the form of Equation 8).
>
> A3: In Sec. 4.2, the subscript ( k ) in Eq. (6) indicates the number of subgoals ({$s_g$}_k means a set), so a complete task (s, g) is divided into ( k+1 ) segments, which we organize into a vector. Eq. (7) calculates its $p$-norm, and Eq. (8) minimizes the optimization objective through a self-supervised learning method, where $\hat{s_g}$ represents the sampled subgoal, rather than the output of the high-level policy $\pi^h$. We appreciate your concerns regarding Eq. (8). As you mention, our construction of the self-supervised method indeed leans more towards AWR-like. The difference is that we compare the advantages of sampled subgoals over those output by high-level policies, rather than comparing the advantage of the current policy over the historical (or mixed) policy as in AWR. We revise this wording in the new version.

---

> > ### Author Response · Authors · 2024-11-21
> >
> > Q4: Why does selecting all virtual goals cause the method to fail? (in Fig 7)
> >
> > A4: Your observation is very astute. We believe the reason for this phenomenon is that learning solely from experiences constructed with virtual goals is highly unstable in the early stages of policy learning. At this point, the agent is not yet capable of solving simple tasks (where the start and end points are relatively close), leading to large errors in the value function, which in turn affects subgoal planning. Introducing even a small portion of actual goals (a:v = 0.1:0.9) can result in significant improvement. Therefore, we point out in Sec. 4.3: "We attribute this to actual experiences enabling the agent to quickly master simple tasks, while virutal experiences help the agent tackle more challenging, long-horizon tasks.".
> >
> > Q5: It is common in hindsight GC implementations to (with some proportion) sample a goal at random from the replay buffer. This alleviates a lot of the challenges with otherwise only choosing goals on the same trajectory. Is this being done for the baselines (specifically for SAC+HER?)
> >
> > A5: Some studies have indeed considered using states randomly sampled from the replay buffer at a certain ratio as relabeling goals [2,6]. However, we found that their task settings are limited to a fixed starting point, where both the actual goals and the virtual goals diverge outwards from the fixed starting point. As a result, virtual experiences do not introduce excessive noise, but this can easily happen in tasks with random starting and ending points. Our implementation of "SAC+HER, PIG, HIGL, GCSL" only considers future states from the same trajectory as relabeling goals. And we use the same goal relabeling setup for RIS as our VE (a:v=0.5:0.5).
> >
> > Q6: In Equation 9, it seems that k is being overloaded both as an iteration index (e.g. theta_{k+1}), and also as the subgoal index?
> >
> > A6: Yes, that was a writing oversight on our part, and we apologize for any inconvenience it caused. We appreciate your suggestions regarding the writing details and will make corrections in future versions.
> >
> >
> >
> >
> >
> > [1] Florensa, C., Held, D., Geng, X., & Abbeel, P. Automatic goal generation for reinforcement learning agents. (ICML2018)
> >
> > [2] Pong, V., Dalal, M., Lin, S., Nair, A., Bahl, S., & Levine, S. Skew-Fit: State-Covering Self-Supervised Reinforcement Learning. (ICML2020)
> >
> > [3] Sekar, R., Rybkin, O., Daniilidis, K., Abbeel, P., Hafner, D., & Pathak, D. Planning to explore via self-supervised world models. (ICML2020)
> >
> > [4] Chane-Sane, E., Schmid, C., & Laptev, I. Goal-conditioned reinforcement learning with imagined subgoals. (ICML2021)
> >
> > [5] Kim, J., Seo, Y., Ahn, S., Son, K., & Shin, J. Imitating Graph-Based Planning with Goal-Conditioned Policies. (ICLR2023)
> >
> > [6] Pitis, S., Chan, H., Zhao, S., Stadie, B., & Ba, J. Maximum entropy gain exploration for long horizon multi-goal reinforcement learning. (ICML2020)

---

> ### Author Response · Authors · 2024-11-25
>
> Dear Reviewer TXca
>
> Thanks for your time and comments on our work!
>
> We have tried our best to address the concerns and provided detailed responses to all your comments and questions.
>
> Would you mind checking our response and confirming whether you have any further questions?
>
> Best regards,
> Authors of #5840

---

### Official Review · Reviewer_i5cV · 2024-11-10

**Soundness:** 3
**Presentation:** 2
**Contribution:** 4
**Rating:** 5
**Confidence:** 5

**Summary:**

The paper proposes a variant to hindsight experience replay (HER) methods for GCRL that improves the quality of hindsight-relabeled goals using a high-level subgoal proposal policy, and additionally uses a task density model to produce better curricula of goals to reach. On a few RL benchmarks (AntMaze, Reacher, Sawyer), this method shows substantial improvements over existing GCRL methods.

**Strengths:**

- Lots of interesting ideas presented: hierarchical policy for proposing subgoals for the hindsight experience relabeling; density model for designing good subgoal curricula; imitation regularization objective using subgoal-policy; p-norm for generating subgoal sequences
- The method outperforms prior online GCRL methods on several tasks (AntMaze, Sawyer, Reacher), showing strong performance

**Weaknesses:**

- There are many moving parts to this method which aren't ablated (the "three key ways" mentioned at the start of Sec. 4 seem to be relatively distinct algorithmic decisions that should be ablated).
- I found the method presentation to be confusing, in particular how the various components (like "task-probability-density-based curriculum learning" and "subgoal planning" relate)
- Minor formatting issues: use \citep{} for parenthetical citations, the green citecolor is a bit hard to read

**Questions:**

- Can $1<p<2$ be used in the p-norm for subgoals? When might $p=\infty$ be suboptimal?
- In Fig. 12, most of the proposed subgoals appear very close to the goals
- Is there a theoretical justification for why imitating the subgoal-generating policy in eq. (9) is beneficial?
- Likewise, is there any mathematical statement that can be made for why this goal relabeling approach is superior to HER?
- How exactly is the task density model used for curriculum learning? This seems to be an important component to the method (Fig. 10), but I didn't understand how it fits into, e.g., Eqs. (5) and (7).
- What is the justification for the direction of the KL divergence in Eq. (9)? What is $\pi^{\text{prior}}$?

---

> ### Author Response · Authors · 2024-11-21
>
> We thank you for your reviews and address your concerns as follows:
>
> Q1: Can $1<p<2$ be used in the $p$-norm for subgoals? When might $p=\infty$  be suboptimal?
>
> A1: For the $p$-norm, we primarily focus on the commonly used cases of $p=1,2,\infty$. We did not consider the range $1<p<2$; however, computations for these cases are feasible as long as $f = x^p$ satisfies the condition of being a convex function. In our experience, $p = \infty$ demonstrates broader applicability, but it is not always optimal (in terms of optimizing efficiency). This is because it minimizes the largest term in Eq. (6) at all times. While this approach to solving subgoals is relatively stable, there is likely room for further improvement.
>
> Q2: In Fig. 12, most of the proposed subgoals appear very close to the goals.
>
> A2: We apologize for the misunderstanding in this instance. Fig.12 illustrates the final states achieved by the agent under the guidance of the three methods in the most difficult tasks, visualized through the $(x,y)$ coordinates.
>
> Q3: Is there a theoretical justification for why imitating the subgoal-generating policy in eq. (9) is beneficial?
>
> A3: To the best of our knowledge, the most relevant literature is [1], which discusses the feasibility of persistently imitating advantage-based policies. However, However, we have not encountered theoretical research that is closely related to this paper. We are happy to share some intuitive insights for your consideration.
>
> We regard the additional imitation learning component as a form of regularization. When the agent is capable of achieving the goal, the subgoal-conditioned policy is similar to the goal-conditioned policy. In such cases, the regularization term neither introduces conflict nor imposes excessive constraints. Conversely, when the agent's success rate in achieving subgoals significantly exceeds its success rate for the goal, the subgoals effectively simplify the task. In this cases, the subgoal-conditioned policy is likely to diverge from the goal-conditioned policy, thereby correcting the policy gradient derived solely from the value function $Q(s, \pi(s,g), g)$. We believe this is the primary role played by imitation learning in Eq. (9).
>
> Q4: Likewise, is there any mathematical statement that can be made for why this goal relabeling approach is superior to HER?
>
> A4: The *generalized goal relabeling* method proposed in Sec. 4.1 is an extension of HER. Our primary idea is to incorporate a wider range of potential goals—whether they are actual goals, virtual goals, or artificially crafted expert goals—as relabeling goals. This approach enriches the learning experience for off-policy RL. We have identified a recent study [2] that explains the high sample efficiency of goal relabeling.
>
> Q5: How exactly is the task density model used for curriculum learning? This seems to be an important component to the method (Fig. 10), but I didn't understand how it fits into, e.g., Eqs. (5) and (7).
>
> A5: As described in Sec. 4.1, we employ a task density model to filter virtual goals (see Fig. 2(Left)). This model does not directly participate in the optimization process (e.g. Eqs. (7) and (9)) but rather selects appropriate goals (e.g. 0.8$\bar e$<$e_i(s,g_i)$<1.2$\bar e$) from randomly sampled virtual goals. This filtering process is an integral part of constructing a progressive curriculum for learning.
>
> Q6: What is the justification for the direction of the KL divergence in Eq. (9)? What is $\pi^{prior}$?
>
> A6: The $\pi^{prior}$ referred to Eq. (9) is a generic prior policy, and in this paper, we adopt the policy target network $\pi_{\theta_{m-1}}$ (the paper uses $k$ to denote the number of subgoals, and here we use $m$ to represent the update step) from the soft-update process. In Eq. (9), $\pi^{prior}$ serves as the imitation target, and the KL divergence encourages the current goal-conditioned policy $\pi_{\theta_m} (s,g)$ to approach the subgoal-conditioned policy $\pi^{prior}(s,s_{g_i})$. The rationale for this process is connected to the intuitive assumptions presented in A3.

---

> > ### Author Response · Authors · 2024-11-21
> >
> > Q7: There are many moving parts to this method which aren't ablated (the "three key ways" mentioned at the start of Sec. 4 seem to be relatively distinct algorithmic decisions that should be ablated).
> >
> > A7: In Fig. 7 (left), we examine the impact of the proportion of actual and virtual goals (which indirectly affects curriculum learning). In Fig. 7 (middle), we assess the influence of curriculum learning itself, corresponding to (a) in the "three key ways." Since (b) primarily focuses on subgoal planning, we select "os" (oracle subgoals) and "rs" (random subgoals) as baselines in Fig. 7 (right) to validate the effectiveness of our subgoal planning. As all of our design converge on incorporating the imitation of subgoal-conditioned policies into the policy improvement process, the control group for (c) is considered methods like "SAC+HER," which do not include imitation.
> >
> > Lastly, we sincerely appreciate your feedback regarding the writing issues, and we will make adjustments to the corresponding sections accordingly.
> >
> >
> >
> > [1] Oh, J., Guo, Y., Singh, S., & Lee, H. Self-imitation learning. (ICML2018)
> >
> > [2] Zheng, S., Bai, C., Yang, Z., & Wang, Z. How Does Goal Relabeling Improve Sample Efficiency?. (ICML2024)

---

> ### Author Response · Authors · 2024-11-25
>
> Dear Reviewer i5cV
>
> Thanks for your time and comments on our work!
>
> We have tried our best to address the concerns and provided detailed responses to all your comments and questions.
>
> Would you mind checking our response and confirming whether you have any further questions?
>
> Best regards,
> Authors of #5840

---

> > ### Comment · Reviewer_i5cV · 2024-12-03
> > **Response**
> >
> > Thank you for your detailed response.

---

### Author Response · Authors · 2024-11-21
**General Response**

We thank all reviewers for their valuable comments. We have summarized the main points as follows:

### strengths

* The results are very thorough (TXca, 6ueC)
* Propose a concise and effective subgoal planning method (i5cV, TXca, SKRR)
* Visualizations to aid understanding (TXca, 6ueC)
* The improvement over the baseline is significant (i5cV, TXca, SKRR, 6ueC)

### weaknesses

* The statement of the methods in Sec. 4.1 needs improvement to make it easier to understand (SKRR, 6ueC)
* How is curriculum learning set up and how does it work? (i5cV, TXca, 6ueC)
* Some minor shortcomings in writing (i5cV, TXca, SKRR)

We sincerely thank the reviewers for acknowledging the contributions of this paper. We provide a unified response to some common issues here, hoping to help everyone clearly understand the ideas presented in this paper.

The motivation of this paper is how to efficiently utilize historical experience. Inspired by HER, we believe that virtual experience also contains rich knowledge (derived from different tasks $(s,g)$), and we attempt to capture these knowledge by planning subgoals and imitating policies conditioned on these subgoals. The framework we propose is concise, and all designs ultimately focus on the imitation learning term of policy improvement, which can be regarded as a regularization term of the traditional value-based method. Furthermore, we want to emphasize the potential benefits of fully utilizing historical experience. For example, in Fig. 8, we enabled the agent to learn tasks it has never experienced in a constrained environment. We are confident that this approach holds promise for developing in the context of online and offline RL.

In contrast, curriculum learning is an independent module primarily aimed at providing a step-by-step virtual goal curriculum to enhance the efficiency and stability of learning. However, it must be acknowledged that the currently proposed task probability density model does not perform well in all environments (as shown in Figure 5(b), where the performance is inferior to (a)). Moreover, the method used is relatively rudimentary, such as directly selecting virtual goals that meet the conditions from a fixed range as relabeling goals. Of course, we believe that it can also benefit from better designs in the future.

We greatly appreciate the reviewers' comments regarding the writing shortcomings and are committed to addressing them by clarifying our methods in the revised version. For a more detailed response, please refer to the official comments below.

---

> ### Author Response · Authors · 2024-11-27
> **General Response**
>
> We thank all reviewers for their valuable comments. We have revised the paper, according to their comments. All revised parts are marked as blue. The main revisions are summarized as follows:
> 1. We have corrected some mathematical issues pointed out by reviewers.
> 2. We highlight the differences between VE and existing methods in related work.
> 3. We rewrote Section 4.1 to clarify that task probability density is used to screen suitable virtual goals, which is part of constructing training data and is unrelated to subgoal planning.
> 4. We provide additional details on tuning hyperparameters in the Appendix H.

---

### Public Comment · ~Xing_Lei1 · 2025-03-28
**Looking forward to working with you in the field of goal-conditioned rl**

Dear Author,

I hope this comment is good for you. I recently came across your latest article and found it highly relevant to my research. As a researcher in goal-conditioned reinforcement learning, I am particularly interested in your work and its potential applications.

If possible, could you please leave your contact information so that we can communicate? My email is leixing@stu.xjtu.edu.cn. I am currently working on targeted conditional reinforcement learning.

Looking forward to your reply. Thank you for your time and consideration!

Best regards,

Xing Lei

---

### Meta-Review · Area_Chair_AeKS · 2024-12-21

**Metareview:**

This paper proposes a method for goal-conditioned RL that learns with a curriculum of relabelled goals and distills learned behavior for reaching s --> subgoal into a model for reaching s --> goal. On a few RL benchmarks (AntMaze, Reacher, Sawyer), this method shows substantial improvements over existing GCRL methods.

Reviewers appreciated the "elegance" of the core idea (distilling policies from a subgoal to a further goal) and thought the empirical results were thorough and strong, including the ablation experiments. Reviewers appreciated both the problem setup, the inclusion of experimental details, and the visualizations in the paper.

Some reviewers found the presentation confusing (e.g., how the components interact, Sec. 4.2, symbols in formulas). One reviewer questioned the inclusion of the curriculum, which adds complexity without much boost in performance. The tasks used for evaluation are primarily navigation tasks, so it remains unclear whether the method scales to tasks with higher (intrinsic) dimension (e.g., manipulation).

Taken together, I think the core idea is strong but that the writing and experiments need to be improved (see suggestions in the reviews) before publications. I therefore recommend that the paper be rejected.

**Additional Comments On Reviewer Discussion:**

While reviewers thought that core idea was strong, they believed that some sections of the paper needed significant rewriting, even after the authors had made some revisions. The reviewers also noted that some of the evaluation decisions seem odd and that important experimental details are missing. Authors acknowledge that scaling to higher-dimensional tasks is an important area for future work, as is figuring out how to simplify the proposed mechanism for curriculum learning.

---

### Decision · Program_Chairs · 2025-01-22

Reject